# CaMKII autophosphorylation can occur between holoenzymes without subunit exchange

Iva Lučić[1,2]*, Léonie Héluin[1,2], Pin-Lian Jiang[2], Alejandro G Castro Scalise[1,2], Cong Wang[2], Andreas Franz[3], Florian Heyd[3], Markus C Wahl[3,4], Fan Liu[2,5], Andrew JR Plested[1,2,6]*

[1]Institute of Biology, Cellular Biophysics, Humboldt Universität zu Berlin, Berlin, Germany; [2]Leibniz-Forschungsinstitut für Molekulare Pharmakologie, Berlin, Germany; [3]Institute of Chemistry and Biochemistry, Freie Universität Berlin, Berlin, Germany; [4]Helmholtz-Zentrum Berlin für Materialien und Energie, Macromolecular Crystallography, Berlin, Germany; [5]Charité-Universitätsmedizin Berlin, Berlin, Germany; [6]NeuroCure, Charité Universitätsmedizin, Berlin, Germany

*For correspondence:
iva.lucic@hu-berlin.de (IL);
andrew.plested@hu-berlin.de
(AJRP)

**Competing interest:** The authors declare that no competing interests exist.

**Abstract** The dodecameric protein kinase CaMKII is expressed throughout the body. The alpha isoform is responsible for synaptic plasticity and participates in memory through its phosphorylation of synaptic proteins. Its elaborate subunit organization and propensity for autophosphorylation allow it to preserve neuronal plasticity across space and time. The prevailing hypothesis for the spread of CaMKII activity, involving shuffling of subunits between activated and naive holoenzymes, is broadly termed subunit exchange. In contrast to the expectations of previous work, we found little evidence for subunit exchange upon activation, and no effect of restraining subunits to their parent holoenzymes. Rather, mass photometry, crosslinking mass spectrometry, single molecule TIRF microscopy and biochemical assays identify inter-holoenzyme phosphorylation (IHP) as the mechanism for spreading phosphorylation. The transient, activity-dependent formation of groups of holoenzymes is well suited to the speed of neuronal activity. Our results place fundamental limits on the activation mechanism of this kinase.

## Editor's evaluation

This manuscript reports the fundamental finding that an oligomeric protein kinase, CaMKII, can be phosphorylated by another molecule of the holoenzyme in a manner that does not involve subunit exchange. The evidence for the main conclusion is compelling, and supported by several independent experiments. If independently confirmed in the future, the study will stand as having provided a novel regulatory mechanism for the autophosphorylation of this kinase. The work will be of broad interest to molecular and cellular neuroscientists as well as biochemists.

## Introduction

The Calcium-calmodulin-dependent protein kinase 2 (CaMKII) is encoded by 4 genes CaMKII *A*, *B*, *G*, and *D* (*Tombes et al., 2003*). Whilst the γ and δ protein isoforms are expressed throughout the body, CaMKIIα and β show brain-specific expression where they are amongst the most abundant proteins in the neuronal cytosol (*Cheng et al., 2006*; *Sheng and Kim, 2011*). CaMKII is an oligomeric structure of mostly 12 or 14 subunits (*Rosenberg et al., 2006*; *Chao et al., 2011*; *Bhattacharyya et al., 2016*; *Buonarati et al., 2021*). The individual subunits associate into holoenzymes through avid hub

domain interactions. The hub connects to the N-terminal kinase domains through a variable-length linker and regulatory domain. The peripheral location of the kinase domains presumably facilitates interaction with each other and substrates (*Chao et al., 2011*; *Myers et al., 2017*; *Buonarati et al., 2021*, *Figure 1A*).

In basal conditions, CaMKII is kept silent because the regulatory domain blocks access to the substrate binding cleft (*Chao et al., 2011*; *Myers et al., 2017*). A canonical form of synaptic plasticity relates to the recruitment of active CaMKII to synapses following calcium (Ca$^{2+}$) influx via the N-methyl-D-aspartate (NMDA)-type glutamate receptors (*Lisman et al., 2002*; *Lisman et al., 2012*). Binding of Ca$^{2+}$-bound calmodulin to the regulatory domain activates CaMKII by liberating its substrate binding site. Auto-phosphorylation at T286 (CaMKIIα numbering) in the regulatory domain prevents rebinding of the regulatory domain back to the kinase and allows substrate engagement. At this stage CaMKII no longer needs Ca$^{2+}$:CaM in order to be active. This mode of operation, which allows CaMKII to remain active after the Ca$^{2+}$ signal (and the associated Calmodulin binding) has finished, is referred to as autonomous activity (*Lou et al., 1986*; *Miller and Kennedy, 1986*; *Lou and Schulman, 1989*; *Waldmann et al., 1990*; *Chang et al., 2019*, *Figure 1B*). Auto-phosphorylation at two other sites in the regulatory domain (T305 and T306) prevents further binding of Ca$^{2+}$:CaM (*Lou and Schulman, 1989*; *Elgersma et al., 2002*). Autonomous activity, was first envisaged as a desirable general mechanism for memory storage by Crick before CaMKII was identified (*Crick, 1984*). Indeed, it was recently concluded, based on the survival of autophosphorylation, that CaMKII activity can outlive its protein turnover time in mouse cultured brain slices, providing important evidence for spread of CaMKII activity to newly synthesized, naïve CaMKII (*Lee et al., 2022*). The oligomeric structure of CaMKII might aid autonomous activity. In particular, the intra-holoenzyme autophosphorylation might provide a reservoir against phosphatase attack, because any dephosphorylated subunits can be 'revitalized' by their neighbors. CaMKIIα organizes a range of functions related to the plasticity of neuronal structure and function. CaMKII is concentrated in dendritic spines, where it bundles actin and phosphorylates its targets such as the α-amino-3-hydroxy-5-methyl-4-isoxazolepropionic acid (AMPA) receptor auxiliary proteins (*Coultrap and Bayer, 2012*; *Herring and Nicoll, 2016*; *Bayer and Schulman, 2019*).

A further key concept, that CaMKII phosphorylation is spread throughout a naïve population of holoenzymes by physical exchange of subunits (i.e. active subunits are swapped into naïve holoenzymes, which then auto-phosphorylate around the ring) is grounded on several observations. First, early reports suggested that naïve holoenzymes do not get phosphorylated when mixed with activated holoenzymes (*Miller and Kennedy, 1986*; *Hanson et al., 1994*; *Rich and Schulman, 1998*; *Chao et al., 2011*), and that the development of autonomous activity is independent of the concentration of holoenzymes (*Bradshaw et al., 2002*). However, these experiments were mostly carried out at low temperature and with short mixing times and at low concentrations that do not mimic the physiological realm of neurons. Second, labeled subunits accumulate into clusters upon activation, as observed in total internal reflection fluorescence (TIRF) microscopy, but the limited resolution of optical microscopy means that clusters could contain multiple holoenzymes (*Stratton et al., 2014*; *Bhattacharyya et al., 2016*). Third, peptides derived from the regulatory domain can selectively break up hubs during electrospray mass spectrometry, but not in solution (*Karandur et al., 2020*).

In this work, we provide data that support a new model in which inter-holoenzyme phosphorylation (IHP) is the predominant mechanism by which CaMKII activity propagates. Subunit mixing experiments, using a panel of approaches and labels, confirm that restraining subunits through the hub domain has no effect on the accumulation of phosphorylation, that holoenzymes reversibly accumulate into clusters upon activation, and that whilst kinase domains interact promiscuously during autophosphorylation, the hub domains do not mix.

## Results
### CaMKII subunits from different holoenzymes can trans-autophosphorylate

Seeking to validate the subunit exchange as a mechanism for spread of kinase activity in CaMKII, we first tested the ability of wild-type CaMKII (CaMKII$^{WT}$) to phosphorylate kinase-dead CaMKII (K42R and D135N double mutant; CaMKII$^{KD}$) in vitro using purified proteins (*Figure 1C and D*; *Figure 1—figure*

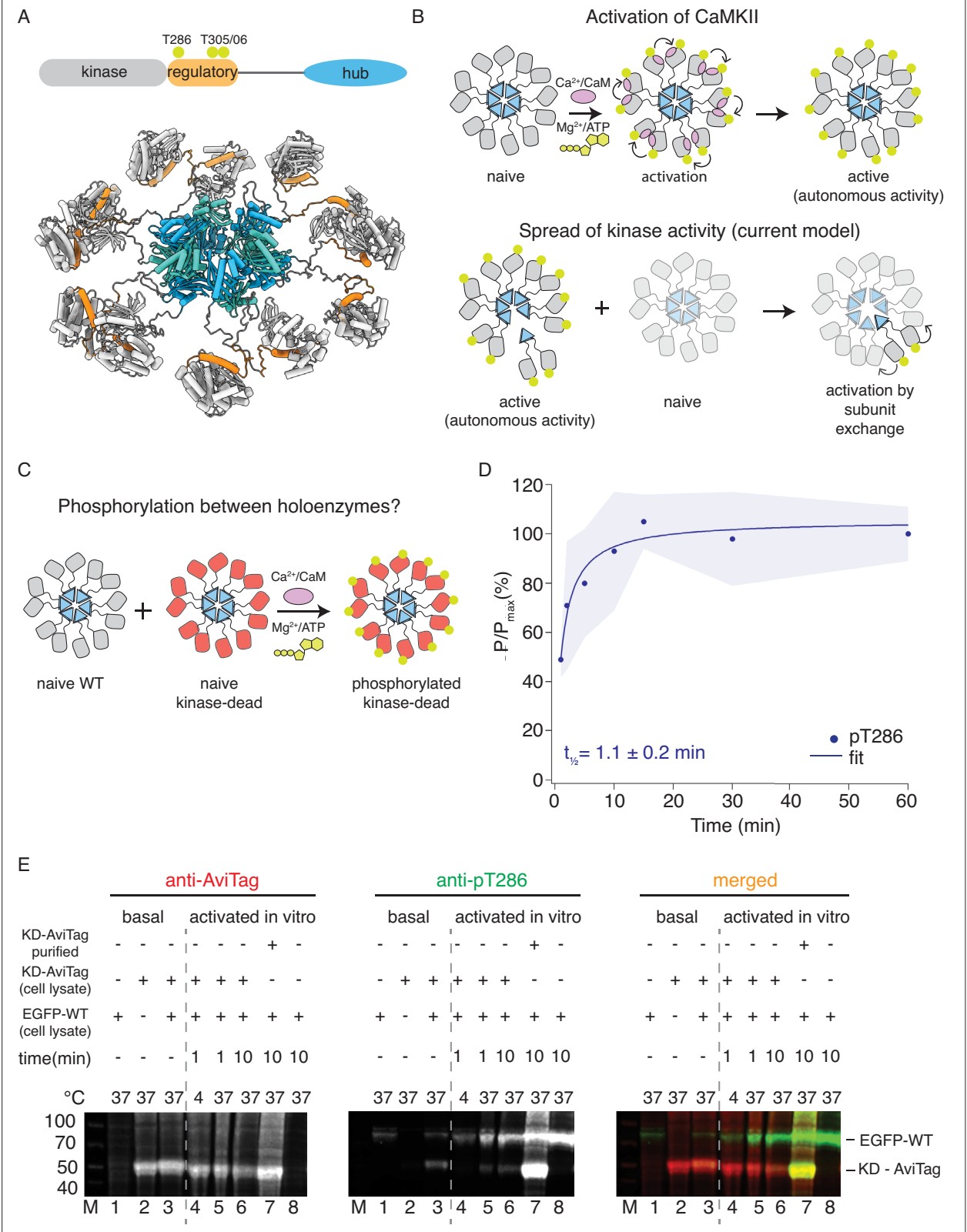

**Figure 1.** CaMKII[WT] phosphorylates CaMKII[KD]. (**A**) Schematic representation of CaMKIIα domain arrangement (top). Yellow circles indicate phosphorylation sites. CaMKIIα holoenzyme structure (bottom, PDB:5u6y). (**B**) Cartoon representation of activation of CaMKII by calcium:calmodulin (top) and proposed mechanism for spread of kinase activity (bottom). Yellow circles indicate phosphorylation sites. (**C**) Schematic representation of experiment performed in panel (**D**). (**D**) Kinase activity of CaMKII[WT] (10 nM) against CaMKII[KD] (4 μM). Half-time of maximum phosphorylation ($t_{1/2}$=1.1 ±

*Figure 1 continued on next page*

*Figure 1 continued*

0.2 min) determined by western blot, using an antibody against pT286. These data were fit with a single component, see *Figure 1—figure supplement 3* for a fit to a repeated experiment with two components. The shading represents standard deviation (SD) from the mean value of three technical replicates. (**E**) Kinase activity of EGFP-CaMKII$^{WT}$ and CaMKII$^{KD}$-AviTag in HEK cells and HEK cell lysates. 'Basal': activity in cells; 'activated in vitro': activity in cell lysates supplemented with 1 µM purified Ca$^{2+}$:CaM and 0.5 mM ATP:Mg$^{2+}$. Lane 1 – autophosphorylation of EGFP-CaMKII$^{WT}$ in singly transfected cells, lane 2 – autophosphorylation of CaMKII$^{KD}$-AviTag in singly transfected cells, lane 3 – autophosphorylation of EGFP-CaMKII$^{WT}$ and CaMKII$^{KD}$-AviTag in co-transfected cells, lane 4 – stimulated kinase activity of EGFP-CaMKII$^{WT}$ from lysate in lane 1 and CaMKII$^{KD}$-AviTag from lysate in lane 2, incubated together at 4 °C for 1 min, lane 5 - stimulated kinase activity of EGFP-CaMKII$^{WT}$ from lysate in lane 1 and CaMKII$^{KD}$-AviTag from lysate in lane 2, incubated together at 37 °C for 1 min, lane 6 - stimulated kinase activity of EGFP-CaMKII$^{WT}$ from lysate in lane 1 and CaMKII$^{KD}$-AviTag from lysate in lane 2, incubated together at 37 °C for 10 min, lane 7 - stimulated kinase activity of EGFP-CaMKII$^{WT}$ from lysate in lane 1 and 8 µM purified CaMKII$^{KD}$-AviTag, incubated together at 37 °C for 10 min, lane 8 - stimulated kinase activity of EGFP-CaMKII$^{WT}$ from lysate in lane 1 37 °C for 10 min.

The online version of this article includes the following source data and figure supplement(s) for figure 1:

**Source data 1.** Uncropped blots used in panel *Figure 1E*.

**Figure supplement 1.** Western blot detection of CaMKII$^{KD}$ phosphorylation by CaMKII$^{WT}$.

**Figure supplement 1—source data 1.** Uncropped blots used for constructing the curves in *Figure 1D* and *Figure 1—figure supplement 1E*.

**Figure supplement 2.** Radioactivity detection of CaMKII$^{KD}$ phosphorylation by CaMKII$^{WT}$.

**Figure supplement 2—source data 1.** Uncropped gels used for constructing the curves in *Figure 1—figure supplement 2C*.

**Figure supplement 3.** Phosphorylation of CaMKII$^{KD}$ by CaMKII$^{WT}$ is concentration-dependent.

**Figure supplement 3—source data 1.** Uncropped blots of examples of immunoblotting used to construct curves in *Figure 1—figure supplement 3A–C*.

**Figure supplement 4.** Phosphorylation of CaMKII$^{KD}$ by 0.5 nM CaMKII$^{WT}$.

**Figure supplement 4—source data 1.** Uncropped blots used for constructing the curves in *Figure 1—figure supplement 3A–C* for 0.5 nM CaMKII$^{WT}$.

**Figure supplement 5.** Phosphorylation of CaMKII$^{KD}$ by 2 nM CaMKII$^{WT}$.

**Figure supplement 5—source data 1.** Uncropped blots used for constructing the curves in *Figure 1—figure supplement 3A–C* for 2 nM CaMKII$^{WT}$.

**Figure supplement 6.** Phosphorylation of CaMKII$^{KD}$ by 10 nM CaMKII$^{WT}$.

**Figure supplement 6—source data 1.** Uncropped blots used for constructing the curves in *Figure 1—figure supplement 3A–C* for 10 nM CaMKII$^{WT}$.

**Figure supplement 7.** Phosphorylation of CaMKII$^{KD}$ by 100 nM CaMKII$^{WT}$.

**Figure supplement 7—source data 1.** Uncropped blots used for constructing the curves in *Figure 1—figure supplement 3A–C* for 100 nM CaMKII$^{WT}$.

supplements 1 and 2). We used CaMKII$^{KD}$ double mutant, because in our hands, the single mutations did not completely abolish kinase activity.

We readily detected phosphorylation of CaMKII$^{KD}$ by either immunoblotting (*Figure 1D* and *Figure 1—figure supplement 1A, B*) against phosphorylated T286, or incorporation of radioactive $^{32}$P (*Figure 1—figure supplement 2*). In these experiments we used 10 nM CaMKII$^{WT}$ and 4 µM CaMKII$^{KD}$, (400-fold excess of KD) in the presence of Ca$^{2+}$:CaM and ATP:Mg$^{2+}$. The WT and KD proteins are the same size (monomer Mw = 55 kDa), which forbids their separation on SDS-polyacrylamide gel electrophoresis (PAGE). However, at 10 nM CaMKII$^{WT}$ phosphorylation was undetectable on a gel (*Figure 1—figure supplement 1D* and *Figure 1—figure supplement 6*), meaning that the phosphorylation signal we detected is overwhelmingly from the 400-fold more abundant kinase-dead protein. The half-times of maximal phosphorylation are different, being 1.1 min for the immunoblotting (*Figure 1D* and *Figure 1—figure supplement 1E*) and 4 min for the P$^{32}$ incorporation (*Figure 1—figure supplement 2C*), perhaps because T286 is phosphorylated early compared to the entire cohort of phosphosites incorporating $^{32}$P.

The ability of CaMKII$^{WT}$ to phosphorylate CaMKII$^{KD}$ is in contrast to previous reports, which failed to detect any inter-holoenzyme phosphorylation of CaMKII$^{WT}$ or CaMKII$^{KD}$ (*Miller and Kennedy, 1986*; *Hanson et al., 1994*; *Rich and Schulman, 1998*). Most previous work used either short incubation time (30–60 s) or low incubation temperature (4 °C) or both; these likely prevented full extent of CaMKII activity. Another difficulty could be the lower expression of CaMKII$^{KD}$ in cell lines, compared to dendritic spines where CaMKII concentration is estimated to be between 20 and 100 µM (*Otmakhov and Lisman, 2012*). This discrepancy means that in vitro experiments, which allowed us to use 4–8 µM CaMKII$^{KD}$ protein, are more representative of the crowded environment of neurons.

Next, we tested phosphorylation of CaMKII in HEK cells and in HEK cell lysates (*Figure 1E*). We expressed EGFP-tagged CaMKII$^{WT}$ alone, CaMKII$^{KD}$-AviTag alone, or both proteins together, and

sought to detect phosphorylation on T286 by western blotting. When expressed separately, we could observe phosphorylation on CaMKII[WT] but not on CaMKII[KD] (*Figure 1E*, lanes 1 and 2). We could, however, detect phosphorylation on CaMKII[KD] when it was co-expressed with EGFP- CaMKII[WT] (*Figure 1E*, lane 3). A confounding explanation for this is that WT and KD subunits might assemble into common holoenzymes during co-expression in cells, allowing KD subunits to be auto-phosphorylated in an intra-holoenzyme context (*Hanson et al., 1994*). Next, we incubated HEK cell lysate expressing EGFP-tagged CaMKII[WT] alone with HEK cell lysate expressing CaMKII[KD]-AviTag alone, in the presence of of 0.5 µM purified Calmodulin as well as 0.5 mM ATP: $Mg^{2+}$ (*Figure 1E*, lanes 4–6), over different incubation temperatures and times. By incubating lysates with separately expressed WT and KD proteins, we wanted to exclude the possibility of the two proteins assembling into the same holoenzymes during translation, which is to be expected when they are co-expressed. Similar to previous reports (*Hanson et al., 1994*; *Rich and Schulman, 1998*) we failed to detect phosphorylation on CaMKII[KD]-AviTag after 1 min incubation at 4 °C (*Figure 1E*, lane 4). In contrast, the phospho-signal was readily detected on CaMKII[KD]-AviTag at 37 °C (*Figure 1E*, lanes 5 and 6), indicating that temperature and incubation time play an important role in enzymatic activity of CaMKII. Finally, when we incubated HEK cell lysate expressing EGFP-tagged CaMKII[WT] with 4 µM CaMKII[KD]-AviTag purified from *E. coli*, we could detect robust phosphorylation on CaMKII[KD] protein (*Figure 1E*, lane 7), indicating the importance of CaMKII concentration in this process. Having in mind high dendritic CaMKII concentrations, this experiment indicates that crowding of CaMKII holoenzymes, rather than subunit exchange per se, might be important for spreading phosphorylation.

Another reason why inter-holoenzyme phosphorylation was previously neglected is due to the evidence for concentration-independent autophosphorylation of CaMKII[WT] (*Bradshaw et al., 2002*). The premise of Bradshaw and colleagues was that inter-holoenzyme phosphorylation should be concentration-dependent due to limited resource of the substrate/enzyme kinase holoenzymes, and should therefore depend on the initial CaMKII[WT] concentration. On the other hand, intra-holoenzyme phosphorylation should not be limited by protein concentration, because both substrate and enzyme kinase would belong to the same holoenzyme and therefore their supply would be 'limitless'. They found that autonomous activity of CaMKII[WT] against a peptide-substrate autocamtide-2 was indistinguishable between CaMKII[WT] samples of different concentrations. There are two caveats in this experiment. First, they did not measure directly the trans-autophosphorylation between holoenzymes, but rather the autonomous activity towards the peptide substrate, used as a proxy for directionality of the trans-autophosphorylation (intra- vs. inter-holoenzyme). Second, in the autonomous activity experiments, after activating kinases (by trans-autophosphorylation) at different CaMKII[WT] concentrations, they reduced CaMKII[WT] concentration (from all reactions) to 10 nM and assayed activity with the peptide-substrate, in the end comparing the autonomous activity of the same amount of CaMKII[WT]. To look more directly into the concentration-dependence of CaMKII trans-autophosphorylation, we performed a set of redesigned experiments to directly measure only inter-holoenzyme phosphorylation. We measured phosphorylation of inactive substrate kinase (CaMKII[KD] at 4 µM), by different concentrations of the enzyme kinase (CaMKII[WT]; 0.5 nM, 2 nM, 10 nM, and 100 nM). We found that phosphorylation on T286 of CaMKII[KD] is highly-dependent on CaMKII[WT] concentration: more enzyme kinase (CaMKII[WT]) speeds up phosphorylation on substrate kinase (CaMKII[KD]) (*Figure 1—figure supplements 3–7*). Intriguingly, fitting the data globally revealed a two-step mechanism, with both concentration-dependent and -independent components. The results of these experiments clearly fulfil the criterion of concentration-dependent (and therefore inter-holoenzyme) phosphorylation.

## Holoenzyme constituents do not rearrange into other holoenzymes during activation

Having established that active CaMKII readily phosphorylates subunits from naïve holoenzymes, we next used a genetically encoded photocrosslinker to test whether the phosphorylation of CaMKII[KD] could still be detected if active CaMKII is restricted to its respective holoenzymes. If phosphorylation spreads by subunit exchange between holoenzymes, then restraining subunits is expected to abolish, or at least slow down, the phosphorylation of CaMKII[KD]. We designed several CaMKII mutants in which we placed a BzF residue on different positions in the hub domain, using genetic code expansion technology (*Figure 2A*, *Chin et al., 2002*; *Ye et al., 2008*; *Klippenstein et al., 2014*). BzF is an unnatural amino acid which, when exposed to UV light, covalently crosslinks to residues at ~5 Å distant (*Kauer*

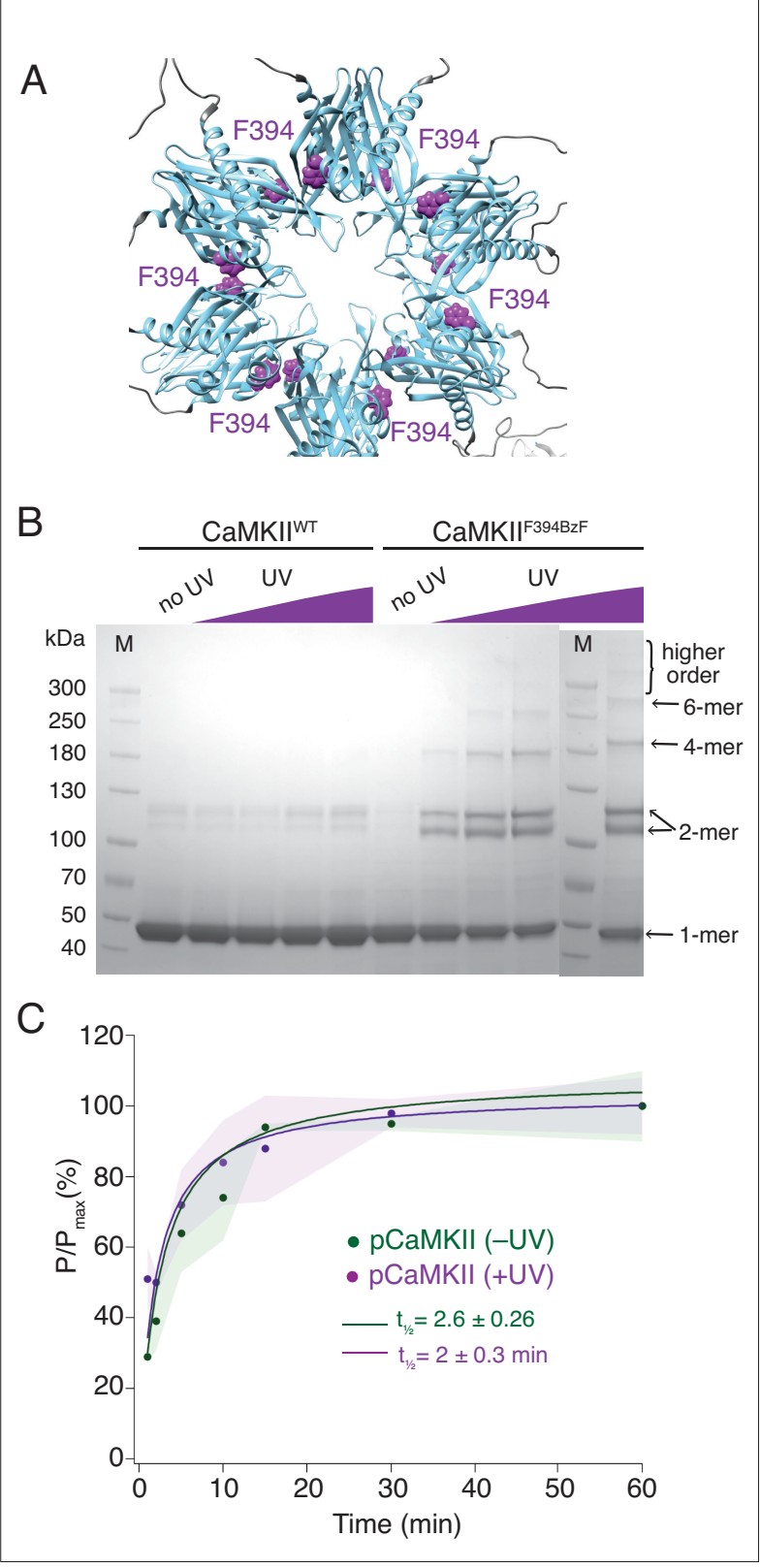

**Figure 2.** Crosslinking CaMKII subunits in the hub domain does not change the rates of trans-autophosphorylation. (**A**) Position of F394 residue (purple) in the hub domain of CaMKIIα (PDB: 5u6y), showing orientation towards the interface between adjacent hub domains within one hub ring. (**B**) Coomassie stained gel of UV-induced crosslinking of CaMKII^F394BzF (right) to higher order oligomers, and CaMKII^WT (left) as a control. Purple

*Figure 2 continued on next page*

*Figure 2 continued*

ramp indicates increasing UV exposure. (**C**) Kinase activity of UV treated (+UV) and untreated (-UV) CaMKII$^{F394BzF}$ against CaMKII$^{KD}$ - AviTag. Phosphorylation was measured as incorporation of radioactive AT$^{32}$P at phosphorylation sites on CaMKII$^{KD}$ -AviTag (pCaMKII). Half-times of maximum phosphorylation are determined to t1⁄2=2.6 ± 0.3 min for untreated and t1⁄2=2 ± 0.3 min for UV treated CaMKII$^{F394BzF}$. The shading represents standard deviation (SD) from the mean value of three technical replicates.

The online version of this article includes the following source data and figure supplement(s) for figure 2:

**Source data 1.** Uncropped Coomassie stained gels used in panel *Figure 2B*.

**Figure supplement 1.** Properties of CaMKII$^{F394BzF}$: kinase activity and size exclusion chromatography.

**Figure supplement 1—source data 1.** Uncropped images of radioactive gels used to construct the curves in *Figure 2C*.

**Figure supplement 2.** Hub domain mutant CaMKII$^{H418BzF}$ phosphorylates CaMKII$^{KD}$ irrespective of crosslinking.

**Figure supplement 2—source data 1.** Uncropped images of Coomassie stained and radioactive gels used to construct the curves in *Figure 2—figure supplement 2B*.

**Figure supplement 3.** Radioactivity detection of substrate phosphorylation by CaMKII$^{F394BzF}$.

**Figure supplement 3—source data 1.** Uncropped images of radioactive gels used in *Figure 2—figure supplement 3*.

*et al., 1986*; *Young et al., 2010*). Exposing hub domain BzF mutants to UV light of 365 nm wavelength generated higher order oligomeric bands on an SDS-PAGE gel (*Figure 2B*). We chose the F394BzF mutation for downstream phosphorylation assays, because this mutant behaved like the WT and KD protein in terms of dodecameric assembly and had similar activity to WT (*Figure 2—figure supplement 1*).

We performed two sets of experiments using CaMKII$^{F394BzF}$ and CaMKII$^{KD}$. First, we monitored the phosphorylation kinetics of CaMKII$^{KD}$ when incubated with CaMKII$^{F394BzF}$, which was either previously treated with UV light or not (*Figure 2C*). We assume that UV treatment crosslinks monomers, imprisoning them within their original holoenzymes and forcing them to phosphorylate CaMKII$^{KD}$ in separate holoenzymes in trans, in order to spread activity. *Figure 2C* shows that the UV treatment of CaMKII$^{F394BzF}$ before the incubation with CaMKII$^{KD}$ and activation stimuli did not slow down the phosphorylation kinetics of CaMKII$^{KD}$. Regardless of whether CaMKII$^{F394BzF}$ was crosslinked or not, phosphorylation of CaMKII$^{KD}$ proceeded with a half time of 2 min. We performed crosslinking of CaMKII$^{F394BzF}$ at concentration of 8 μM, and then diluted it to 10 nM for the phosphorylation assay, whereas we kept the CaMKII$^{KD}$ concentration at 4 μM in the phosphorylation assay, again assuring that the overwhelming majority of the signal from the autoradiography gel is from the kinase-dead protein (*Figure 2—figure supplement 1*). The same lack of effect of UV induced crosslinking was also observed for another BzF hub domain mutant, CaMKII$^{H418BzF}$ (*Figure 2—figure supplement 2*). Finally, restricting the ability of CaMKII$^{F394BzF}$ to dissociate from holoenzymes also failed to alter phosphorylation of the well-known Syntide-2 substrate (*Hashimoto and Soderling, 1987*, *Figure 2—figure supplement 3*).

Since active forms of CaMKII can efficiently phosphorylate CaMKII$^{KD}$ in vitro (*Figures 1D and 2C* and *Figure 2—figure supplement 2*), we used the F394BzF mutant to investigate whether we could capture CaMKII$^{KD}$ subunits with BzF crosslinks during phosphorylation (*Figure 3A*). For this purpose, we incubated 4 μM CaMKII$^{F394BzF}$ with 4 μM CaMKII$^{KD}$-AviTag under activating conditions at 37 °C for 1 h. After this substantial mixing period, and at which point the phosphorylation reaction had already plateaued, we treated the samples with UV light to promote BzF crosslinking. We reasoned that if the subunit exchange were to have occurred during phosphorylation of CaMKII$^{KD}$ by CaMKII$^{F394BzF}$, some fraction of CaMKII$^{KD}$ protein should be captured by UV-induced crosslinking, due to mixing of CaMKII$^{F394BzF}$ and CaMKII$^{KD}$ subunits. The kinase-dead protein had an AviTag on the C-terminus, right after the hub domain. We could detect a faint band of the AviTag signal corresponding to a CaMKII dimer (*Figure 3B*, lane 8) where we had CaMKII$^{KD}$ incubated with CaMKII$^{F394BzF}$ under activation conditions. But critically, this dimeric band upon UV exposure could also be detected in the absence of BzF (lane 10, *Figure 3B*). The faint dimeric band is also present when we treat CaMKII$^{WT}$-AviTag protein alone with UV light (*Figure 3—figure supplement 1B*, lanes 4 and 8). This UV-induced dimerization of non-BzF containing CaMKII is probably mediated via oxidation (*Klippenstein et al., 2014*), since

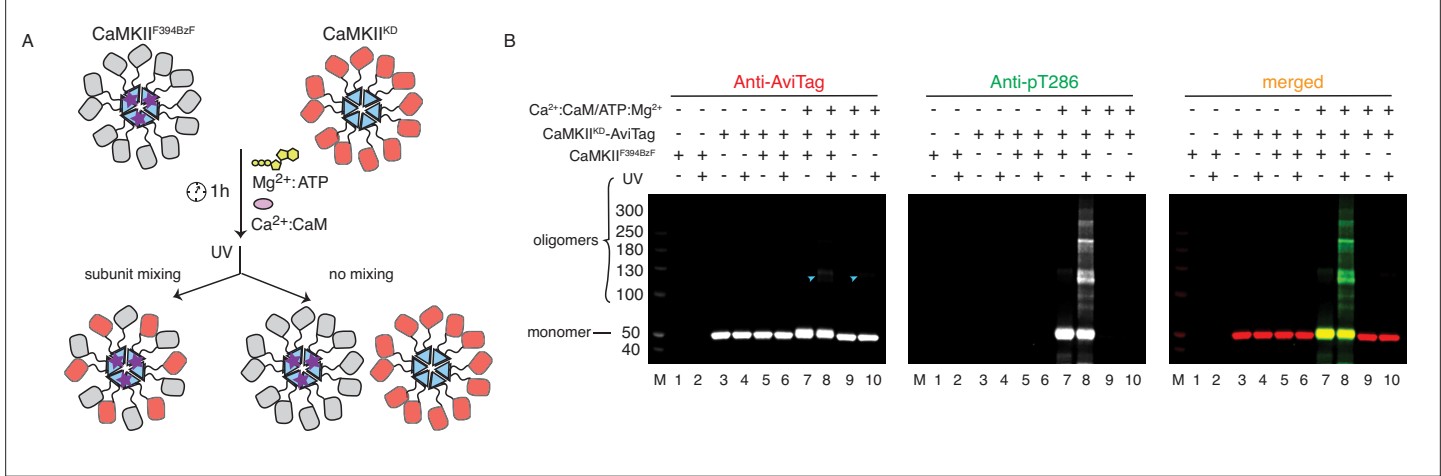

**Figure 3.** CaMKII holoenzymes do not mix during activation. (**A**) Schematic representation of the experiment performed in panel (**B**) and possible outcomes. (**B**) Western blot detection of potential CaMKII^KD-AviTag incorporation in CaMKII^F394BzF holoenzymes. Lane 1 – CaMKII^F394BzF, lane 2 - CaMKII^F394BzF treated with UV, lane 3 - CaMKII^KD-AviTag, lane 4 - CaMKII^KD-AviTag treated with UV, lane 5 - CaMKII^F394BzF incubated with CaMKII^KD-AviTag, lane 6 - CaMKII^F394BzF incubated with CaMKII^KD-AviTag, then UV treated, lane 7 - CaMKII^F394BzF incubated with CaMKII^KD-AviTag and activation stimuli (Ca$^{2+}$:CaM and Mg$^{2+}$:ATP), lane 8 - CaMKII^F394BzF incubated with CaMKII^KD-AviTag and activation stimuli (Ca$^{2+}$:CaM and Mg$^{2+}$:ATP), then UV treated, lane 9 - CaMKII^KD-AviTag incubated with activation stimuli (Ca$^{2+}$:CaM and Mg$^{2+}$:ATP), lane 10 - CaMKII^KD-AviTag incubated with activation stimuli (Ca$^{2+}$:CaM and Mg$^{2+}$:ATP), then UV treated. Blue arrowheads indicate nonspecific, UV-induced, dimerization of CaMKII^KD-AviTag, independent of BzF.

The online version of this article includes the following source data and figure supplement(s) for figure 3:

**Source data 1.** Uncropped blots used in panel *Figure 3B*.

**Figure supplement 1.** CaMKII holoenzymes do not mix during activation.

**Figure supplement 1—source data 1.** Uncropped Coomassie stained gels and blots used in *Figure 3—figure supplement 1*.

it can be blocked by addition of a high concentration of reducing agent TCEP (50 mM, *Figure 3—figure supplement 1C*). Notably, signal for the AviTagged CaMKII^KD protein was absent in a ladder of high molecular weight oligomers generated by covalent crosslinking specific to BzF-containing CaMKII (*Figure 3B*, lane 8, merged blot). This result indicated that although CaMKII^KD can be robustly phosphorylated by CaMKII^F394BzF, these proteins do not mix within the same holoenzymes to an extent that can be detected by western blot. To further ensure that this failure to detect subunit exchange was not an artifact of the KD mutant, we repeated the same experiment using CaMKII^WT-AviTag and CaMKII^F394BzF. Again, no AviTag-labeled protein was enriched in higher order oligomeric bands (other than the background low level of non-specific dimers) on the blot after incubation with CaMKII^F394BzF and subsequent UV-treatment (*Figure 3—figure supplement 1B*).

## Inter-holoenzyme contacts between kinase domains but not hub domains are detected during activation

Although CaMKII can perform phosphorylation of subunits coming from different holoenzymes (*Figures 1D and 2C*, *Figure 1—figure supplements 3–7*, *Figure 2—figure supplement 2*), we failed to detect mixing of subunits during this process, using UV-induced crosslinking of BzF CaMKII mutants (*Figure 3B* and *Figure 3—figure supplement 1B*). Since the prevailing model for spread of kinase activity of CaMKII argues for subunit exchange between holoenzymes (*Stratton et al., 2014*; *Bhattacharyya et al., 2016*), we inspected the mixing of holoenzymes using crosslinking detected by mass-spectrometry. Here, we used CaMKII^WT from *E. coli* grown under regular conditions, and mixed it with CaMKII^WT from *E. coli* grown in $^{15}$N supplemented media. In the latter case, we obtained about 97% $^{15}$N incorporation in CaMKII. After incubation of $^{14}$N-and $^{15}$N-incorporated CaMKII^WT with activation stimuli, we added Disuccinimidyl suberate (DSS) in order to crosslink the proteins in activated states. We also performed this reaction on mixed holoenzymes in basal conditions. At two time points, 30 min and 150 min, the samples were digested and subjected to liquid chromatography mass spectrometry (LC-MS, *Figure 4A*, n=2 replicates). Crosslinked peptides detected from mass spectra were classified according to the ratio (*R*) between the intensities of heterotypic (mixed isotopic, that is I$_{14N:15N}$ and

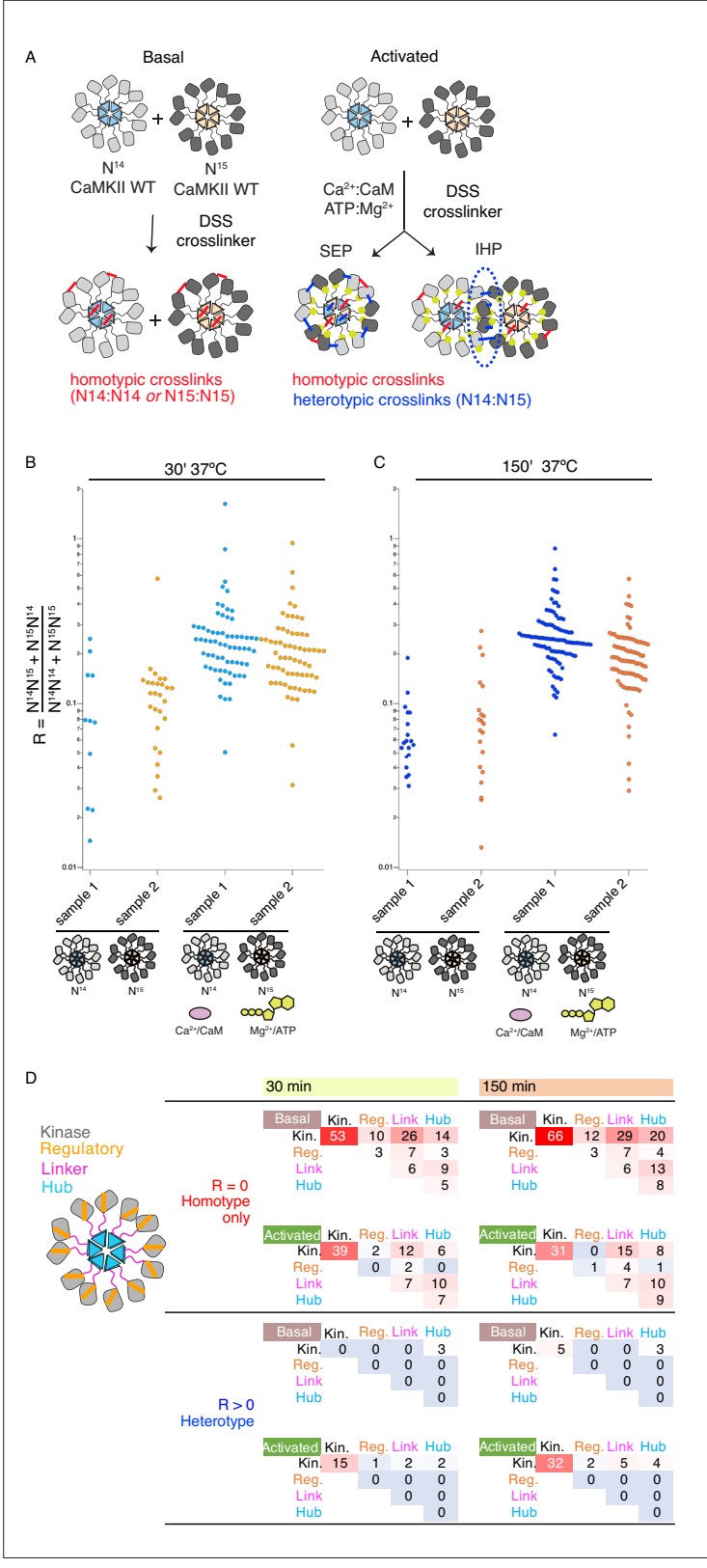

**Figure 4.** Crosslinking mass spectrometry reveals inter-holoenzyme kinase domain contacts during activation. (**A**) Schematic representation of crosslinking experiments and expected outcomes. In the case of subunit exchange, a flat profile of intersubunit crosslinks is expected, whereas for inter-holoenzyme phosphorylation, a bias towards kinase domain crosslinks is expected. (**B**) Heterotypic interactions plotted as R values of two independent

*Figure 4 continued*

replicates in basal and activating conditions. Incubation time was 30 min, prior to addition of DSS crosslinker. (**C**) As for panel (**B**) but with incubation time of 150 min prior to addition of DSS crosslinker. (**D**) Heat map indicating the number of homotypic (upper) and heterotypic (lower) crosslinks by domain under basal and activating conditions, after 30 and 150 min of incubation. Only peptides that were identified in both samples for each condition were counted in the heat map.

The online version of this article includes the following figure supplement(s) for figure 4:

**Figure supplement 1.** XL-MS identifies interactions of CaMKII holoenzymes.

**Figure supplement 2.** Crosslinking sites involving pT286 peptides from mixed isotypes.

I$_{15N:14N}$) and homotypic (uni-isotopic I$_{14N:14N}$ and I$_{15N:15N}$) peaks (see Materials and methods) (*Figure 4—figure supplement 1*). While some crosslinks were only detected between homo-isotopes, others were found (generally with a lower frequency) between both hetero-isotopes and homo-isotopes. The low ratio for heterotypic crosslinks detected is consistent with intra-holoenzyme contacts between subunits being more frequent than contacts between holoenzymes, as expected from local concentration effects.

Across the four conditions, we classified more heterotypic crosslinks after longer incubations (*Figure 4B and C*). Activating conditions increased the number and average ratio of heterotypic crosslinks (*Figure 4B and C*, *Supplementary file 1A and B*) for both 30' and 150' intervals. Next, we counted the number of crosslinks and constructed heatmaps of the crosslinking profiles by domain including only crosslinks found in both replicates (*Figure 4D*). The differential pattern of crosslinks

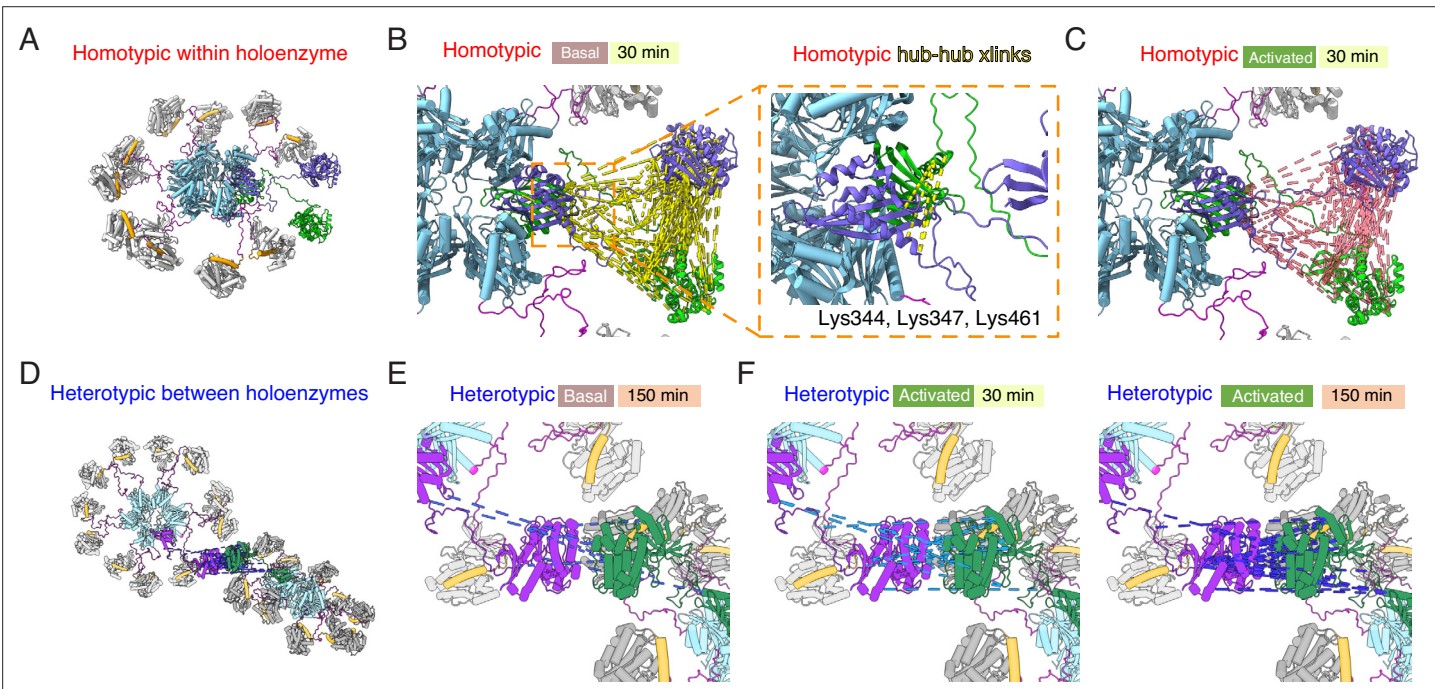

**Figure 5.** Mapping crosslinks onto holoenzyme structure. (**A**) Holoenzyme structure with two neighboring subunits (green and purple) indicated (PDB: 5u6y). Hub domain is in light blue, regulatory segment (docked) in orange. (**B**) Basal crosslinks between homo-isotopes (30 min incubation, 136 crosslinks as dashed yellow lines) plotted onto the holoenzyme as intersubunit interactions. Inset shows the 5 homo-isotopic crosslinks found for interactions between neighboring hub domains. (**C**) Crosslinks between homo-isotopes in activated conditions (30 min, 85 crosslinks as pink dashed lines) showed a similar pattern to the basal condition. (**D**) Two holoenzymes arranged to allow kinase-kinase contacts to form (between green and purple subunits). (**E**) Sparse heterotypic crosslinks (8 in total, blue dashed lines) in the basal condition after 150 min. All heterotypic crosslinks involve the kinase domain. (**F**) In activating conditions, 15 heterotypic kinase-kinase interactions were detected after 30 min (out of 20 total). Over 150 min activation, 32 heterotypic kinase-kinase interactions were found (out of 43 total).

The online version of this article includes the following figure supplement(s) for figure 5:

**Figure supplement 1.** Hypothetical close mode of holoenzyme interaction from MS crosslinks.

between homo- and hetero-isotypes gives insight into the arrangement of subunits. Heterotypic cross-links were restricted to the peripheral kinase domains, and were practically absent in basal conditions (*Figure 4D* and *Supplementary file 1C and D*). In contrast, homotypic crosslinks preferred the kinase domain but otherwise showed a relatively flat profile (*Figure 4D*). Each subdomain (kinase, regulatory segment, linker, and hub) readily linked with the others, consistent with a dynamic holoenzyme (*Figure 4D*). In activating conditions, we detected fewer homotypic crosslinks involving the regulatory domain than in basal conditions, either because CaM binding shields residues or because hetero-geneous phosphorylation hindered peptide identification. It is notable that the regulatory segment was readily crosslinked in basal conditions (23 out of 136 crosslinks involved the regulatory segment), perhaps inconsistent with the canonical view that it is stably docked to the kinase domain in an auto-inhibitory conformation.

Plotting interactions directly onto a model of the CaMKII holoenzyme (*Myers et al., 2017*) gives insight into the phosphorylation pattern (*Figure 5*). Homotypic interactions are relatively uniform and spatially consistent across time points and basal or activated conditions (*Figure 5B and C*). In contrast, heterotypic interactions, modeled onto subunits in a pair of adjacent holoenzymes, are limited to sparse contacts. Without any calculation or docking, a 'close' mode of interaction in which kinase domains from distinct holoenzymes are interdigitated satisfies the distance constraints of crosslinking much better (*Figure 5—figure supplement 1*) without introducing extensive clashes (~100 close atomic clashes mostly in flexible loops, compare with ~90 k atoms per holoenzyme).

Across each of the four conditions, between 5 and 9 homotypic crosslinks were found between hub domains (*Figure 5B*). A frequently-observed crosslink involved lysines at 344 and 347 (*Supplementary file 1A, B, E, F*). Steric considerations favour inter-subunit linkage in this case. On the other hand, heterotypic hub-hub interactions were not detected in any condition, contrary to expectations had subunit exchange between holoenzymes formed of $^{15}N$ and $^{14}N$ subunits occurred (*Supplementary file 1A-H*). The dearth of heterotypic hub-hub crosslinks is in good agreement with absence of UV-induced hub domain crosslinking between subunits from different holoenzymes. In all acti-vated samples, from both homo- and hetero-isotypes, we could detect phosphorylation on T286 (*Figure 4—figure supplement 2*, *Supplementary file 1I and J*), consistent with robust autophosphor-ylation within holoenzymes.

The types of interactions identified here by MS X-linking show that CaMKII holoenzyme is more dynamic than originally thought. The crystal structure of the linker-less inhibited CaMKII holoenzyme (*Chao et al., 2011*) is very compact, and misses important interactions with the linker region. On the other hand, in negative-stain EM structures of the CaMKII holoenzyme with 30 residue linkers (*Myers et al., 2017*) the interaction between the kinase and the hub domain is almost never observed. The MS X-linking analysis also shows that upon activation, the interaction of the hub with the kinase domains is less prominent, with more kinase:kinase interactions presumably corresponding to trans-autophosphorylation both within and between holoenzymes dominating (compare *Supplementary file 1E&F* with G and H).

## Colocalization of CaMKII is activity-dependent and reversible

Much impetus for the subunit exchange theory comes from visualization of fluorescently labeled CaMKII molecules by TIRF microscopy (*Stratton et al., 2014*) or fluorescence resonance energy transfer (FRET) experiments (*Bhattacharyya et al., 2016*). Colocalization of dye-labeled subunits was interpreted to signify that CaMKII can undergo subunit exchange between activated and basal holo-enzymes in order to spread its activity, and that this process was activity-dependent.

We performed TIRF experiments with purified CaMKII$^{WT}$ and/or CaMKII$^{KD}$, which were labeled at their N-termini in vitro using Sortase with Atto-488 or Atto-594 fluorophores (see scheme in *Figure 6A*). In order to monitor colocalization of CaMKII, we first incubated Atto-488-CaMKII$^{WT}$ and Atto-594-CaMKII$^{WT}$, under resting (basal) and activating (activated) conditions (*Figure 6B*). Subunits visualized in the microscope are bound to the coverslip, providing a 'snapshot' of their association at the point of mounting. We used 55 nM CaMKII, a concentration at which dodecamers are around 98% of the population (see below). Limited colocalization between labeled CaMKII$^{WT}$ subunits under resting conditions contrasted with around 50% colocalization of CaMKII$^{WT}$ molecules after incuba-tion in activating conditions (with Ca$^{2+}$:CaM and ATP:Mg$^{2+}$; *Figure 6B and C*). This result is in good agreement with the previous literature (*Stratton et al., 2014*). When we incubated CaMKII$^{WT}$ with

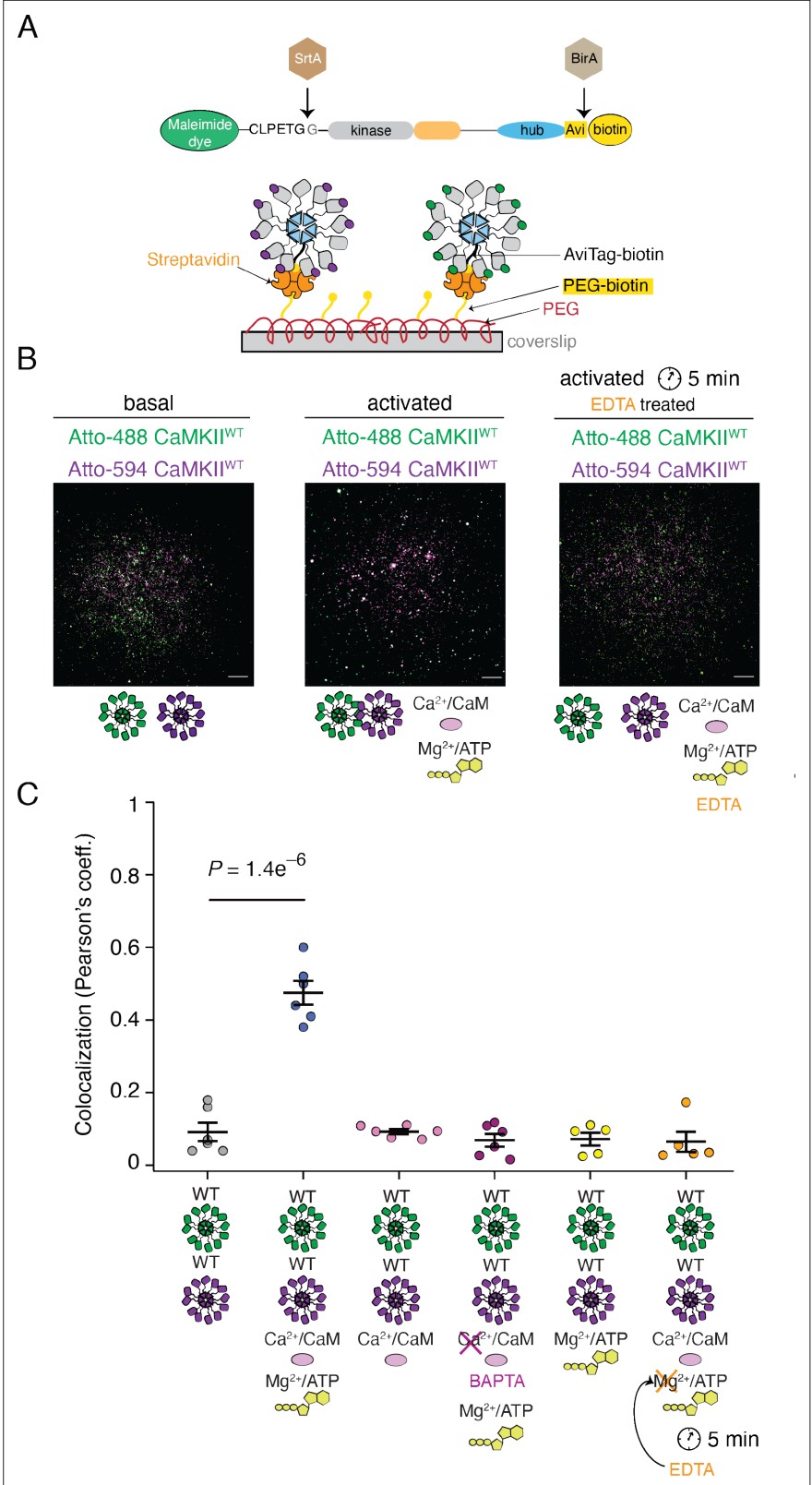

**Figure 6.** Reversible activity-dependent colocalization of CaMKII holoenzymes. (**A**) Schematic representation of CaMKII in vitro enzymatic labeling with maleimide dyes and biotin, and TIRF experimental set-up. (**B**) Representative TIRF images of unactivated (basal) CaMKII[WT] sample (left), activated CaMKII[WT] sample (middle), and CaMKII[WT] sample first activated and then quenched with EDTA (right). Raw, unprocessed, images are shown.

*Figure 6 continued on next page*

*Figure 6 continued*

(**C**) Summary of colocalization analysis (Pearson coefficient) for TIRF images for CaMKII under different conditions. Statistical significance was calculated using a multi-comparison test (Dunnett test) with α=0.05. Mean and standard deviation of the mean are indicated. Each condition was done at least two times on different days, with different protein preps. Five ROIs coming from different slides were selected for analysis for each condition. Scale bar: 10 μm.

The online version of this article includes the following figure supplement(s) for figure 6:

**Figure supplement 1.** Activation stimuli are necessary for CaMKII$^{WT}$ holoenzyme colocalization.

**Figure supplement 2.** CaMKII$^{KD}$ holoenzymes cluster only in the presence of activated CaMKII$^{WT}$ holoenzymes.

ATP:Mg$^{2+}$ but without Ca$^{2+}$:CaM, we could not detect colocalization (***Figure 6C***, ***Figure 6—figure supplement 1C***). The same effect was observed when we stripped the Ca$^{2+}$ ions by incubating Ca$^{2+}$ / CaM with BAPTA, a robust Ca$^{2+}$ -ion chelator (***Collatz et al., 1997***), prior to adding it to the incubation reaction with CaMKII and ATP/Mg$^{2+}$ (***Figure 6C***, ***Figure 6—figure supplement 1B***). Furthermore, incubation of CaMKII$^{WT}$ with Ca$^{2+}$:CaM without ATP:Mg$^{2+}$ again failed to drive CaMKII$^{WT}$ colocalization and confirmed that both activating stimuli are necessary to drive colocalization of CaMKII (***Figure 6C***, ***Figure 6—figure supplement 1A***).

Finally, we looked for a way to initiate the kinase reaction, and putative subunit exchange, and to then quench the kinase reaction and trans-autophosphorylation. We envisaged that if the subunit exchange is indeed taking place during activation, at least partial colocalization of differently labeled CaMKII$^{WT}$ molecules would be observed even after stopping the kinase reaction, thanks to subunit exchange before the reaction has been stopped. To test this possibility, we incubated differentially-labeled CaMKII$^{WT}$ with activating stimuli (both Ca$^{2+}$:CaM and ATP:Mg$^{2+}$), allowed autophosphorylation to occur, and then added EDTA to chelate Mg$^{2+}$ and prevent further phospho-transfer during trans-autophosphorylation (***Rudolf et al., 2014***). In this condition, we surprisingly found a total absence of subunit colocalization between differentially labeled CaMKII$^{WT}$ (***Figure 6C***, ***Figure 6—figure supplement 1D***). This critical result indicates that colocalization is activity dependent but reversible. It is likely not a consequence of the subunits being exchanged between holoenzymes. Rather, what we observe as a colocalization under TIRF comes from different holoenzymes coming into close proximity to trans-autophosphorylate each other in an inter-holoenzyme manner. When ATP was absent, the holoenzymes did not colocalize because they were not trans-autophosphorylating each other. Finally, when we incubated CaMKII$^{WT}$ with CaMKII$^{KD}$ we could still detect colocalization of CaMKII molecules under activating conditions, whereas colocalization of differentially labeled CaMKII$^{KD}$ was completely absent in both conditions (***Figure 6—figure supplement 2***). The colocalization observed between CaMKII$^{WT}$ and CaMKII$^{KD}$ was less pronounced than for mixed WT subunits (***Figure 6—figure supplement 2B***), presumably because half of the sample in this case is inactive, and therefore the activity-driven colocalization occurs to a lesser extent.

## Higher order clusters of holoenzymes detected during activation

If subunits do not exchange, then holoenzymes must come in close proximity in order to trans-autophosphorylate each other. In order to test this hypothesis, we used mass photometry to compare the molecular weight (Mw) of CaMKII$^{WT}$ and CaMKII$^{KD}$ particles under basal and activating conditions. Because mass photometry is a surface-based microscopy method, the micromolar concentrations of CaMKII we used in other assays gave too many particles in the field of view for a valid measurement. The highest CaMKII concentration which we could measure was 400 nM. The high sensitivity of mass photometry gave the substantial advantage that we could work at concentrations of CaMKII particles as low as 10 nM. ***Figure 7A*** shows measured mass distribution of CaMKII$^{WT}$ particles under resting conditions. The only peak which could be fitted with a Gaussian distribution corresponds to particles with Mw corresponding to a dodecamer (592 kDa). This peak was asymmetric on the high molecular weight side, indicating we worked at the high concentration limit of the experiment (***Figure 7—figure supplement 1A***). Upon activation of CaMKII$^{WT}$, we detected two additional sets of particles with molecular weights corresponding to dimer-tetramers and 24–30-mers, in addition to the main dodecameric peak (***Figure 7B***, ***Figure 7—figure supplement 1B***). The measured molecular weights in each peak correspond to CaMKII with bound calmodulin. Peak 0 has a Mw of 157 kDa, which corresponds to CaMKII dimer with two bound calmodulin. Peak 1 corresponds to particles

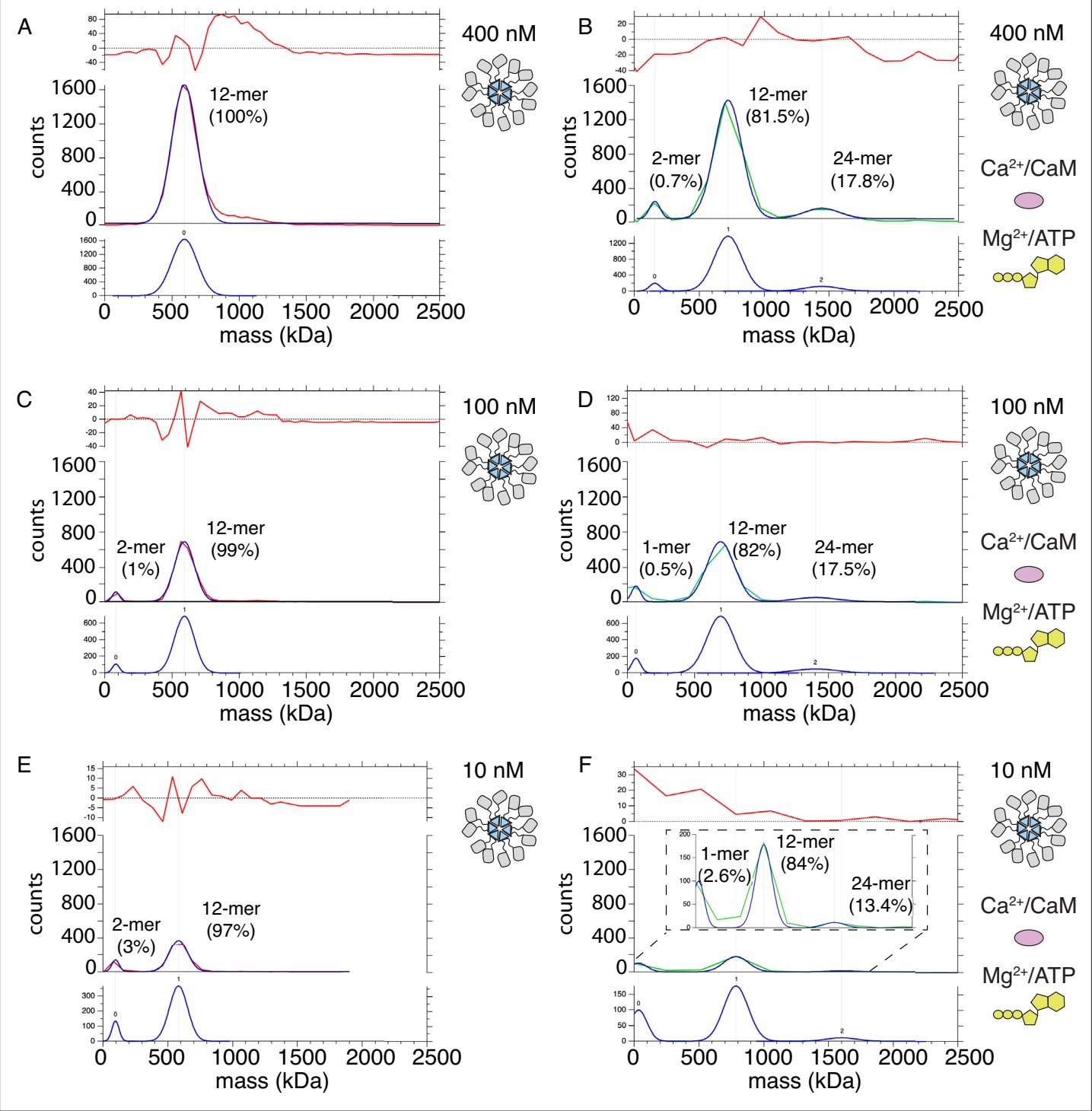

**Figure 7.** Mass photometry detects clusters of holoenzymes forming upon activation. (**A**) Mass distribution of 400 nM CaMKII^WT under basal conditions (red curve). Blue curve is multi-Gaussian fit (shown separately on lower graph). Red curve in upper graph is the fit residual. (**B**) Mass distribution of 400 nM CaMKII^WT under activating conditions (green curve), with multi-Gaussian fit in blue (peaks numbered in lower graph). (**C**) Mass distribution of 100 nM CaMKII^WT under basal conditions (red curve). (**D**) Mass distribution of 100 nM CaMKII^WT under activating conditions (green curve). (**E**) Mass distribution of 10 nM CaMKII^WT under basal conditions (red curve). (**F**) Mass distribution of 10 nM CaMKII^WT under activating conditions (green curve). Inset shows fit on expanded scale.

The online version of this article includes the following source data and figure supplement(s) for figure 7:

**Figure supplement 1.** CaMKII^WT forms higher order clusters during activation.

*Figure 7 continued on next page*

*Figure 7 continued*

**Figure supplement 2.** CaMKII[KD] fails to form higher order clusters during activation.

**Figure supplement 2—source data 1.** Uncropped blots used in *Figure 7—figure supplement 2C*.

with Mw of 721 kDa, close to CaMKII dodecamer with bound calmodulins. Finally, Peak 2 corresponds to particles with Mw of 1442 kDa, which is roughly CaMKII 24-mer with bound calmodulin. We presume that the 24-mer peak corresponds to 2 interacting holoenzymes. The abundance of CaMKII monomer in each peak was 0.7% for CaMKII dimer, 81.5% for CaMKII dodecamer, and 17.8% for CaMKII 24-mer (see "Materials and methods" section for calculation of CaMKII monomer fractions in each peak). Under activating conditions, the majority of CaMKII forms dodecamers, implying that most of trans-autophosphorylation still occurs intra-holoenzyme. However, around 18% of CaMKII forms 24-mers, indicating that the spread of kinase activity might occur between two holoenzymes (inter-holoenzyme). Mass photometry detected less than 1% of CaMKII subunit dimers at 400 nM concentration upon activation, which could be interpreted as either subunit dimers leaving holoenzymes and exchanging with other holoenzymes, or as a general destabilization of a sub-set of holoenzymes brought about by activation. The distribution of Mw peaks of CaMKII at 100 nM concentration showed a monomer-dimer peak (Peak 0=81.5 kDa) already appears in resting conditions (*Figure 7C*; *Figure 7—figure supplement 1C*), albeit at low intensity (representing around 1% of the sample), indicating that lowering CaMKII concentration destabilizes holoenzyme integrity. The main peak still corresponded to dodecameric CaMKII (Peak 1 Mw=591 kDa). Incubation with activating stimuli at 100 nM concentration generated a similar peak distribution as during activation at 400 nM CaMKII. The first peak corresponded to a monomer population (Peak 0 Mw=62 kDa) participating with around 0.5% of CaMKII monomers, the main peak is still dodecameric (Peak 1 Mw=693 kDa) with about 82% of CaMKII monomers participating in this peak, and finally the 24-mer peak was detected (Peak 2 Mw=1404 kDa) consisting of 17.5% of all CaMKII monomers. (*Figure 7D*, *Figure 7—figure supplement 1D*). Lowering CaMKII concentration to 10 nM further destabilized the holoenzyme, as can be seen in the appearance of higher percentage (3%) of particles corresponding to Mw of CaMKII dimer (Peak 0 Mw=95 kDa), but still the majority of particles formed dodecamers (97%, Peak 1 Mw=581 kDa; *Figure 7E*). Remarkably, activation of CaMKII at 10 nM concentration still led to formation of 24-mers (Peak 2 Mw=1600 kDa with 13.4%), dodecameric fraction at 84% (Peak 1 Mw=784 kDa), and monomeric at 2.6% (Peak 0 Mw=33.2 kDa). The peak distribution of activated samples is similar over the range of concentrations, further emphasizing the stability of CaMKII holoenzyme and propensity to form higher order oligomers during activation. CaMKII[KD] particle distributions at 400 nM concentration were similar to the WT protein under basal condition, but the maximum Mw detected was corresponding to decameric holoenzymes (*Figure 7—figure supplement 2A*). Activation of KD protein gave no higher-order peak but generated a small peak of tetrameric CaMKII (12%, *Figure 7—figure supplement 2B*). We assume that the KD protein is capable of binding Calmodulin, since it can only be phosphorylated by the WT protein in the presence of Calmodulin (*Figure 7—figure supplement 2C*). The absence of a higher order oligomeric peak from the CaMKII[KD] sample in activated conditions further emphasizes that reversible CaMKII clustering is driven by trans-autophosphorylation between holoenzymes.

## Discussion

The importance of CaMKII activity, and its idiosyncratic subunit arrangement in dodecameric holoenzymes, have stimulated great interest in its mechanisms of action. Particularly beguiling is the idea that CaMKII can convert brief $Ca^{2+}$ signals corresponding to decisions, calculations and comparisons in neurons into sustained phosphorylation of targets in order to store information in neural circuits (*Lisman et al., 2002*; *Coultrap and Bayer, 2012*; *Tao et al., 2021*). The prevailing model for the requisite spatial and temporal spread of CaMKII autonomous activity argues that CaMKII must exchange subunits between activated and naïve (unactivated) holoenzymes (*Stratton et al., 2014*; *Bhattacharyya et al., 2016*), principally because previous work suggests autophosphorylation between holoenzymes does not occur (*Miller and Kennedy, 1986*; *Hanson et al., 1994*; *Rich and Schulman, 1998*). The data we present here, collected over five separate, complementary techniques, convincingly demonstrate that subunit exchange, if it does occur, is a rare event. We also provide support for a

new model of efficient spread of kinase activity, through inter-holoenzyme trans-autophosphorylation. In this model, CaMKII subunits remain in their respective stable holoenzymes, but phosphorylate neighboring holoenzymes, exploiting the long, flexible linkers which connect the kinase domains to the hub. Contacts between kinase domains are driven by activity, presumably corresponding to multivalent, synergistic autophosphorylation events. The high local concentration of CaMKII in dendrites would be expected to facilitate this interaction between holoenzymes.

Why did previous studies fail to detect inter-holoenzyme phosphorylation? First of all, some studies were not set up to detect it, because measurements were dominated by the fast intra-holoenzyme reaction, which proceeds with terrific speed (*Bradshaw et al., 2002*). In previous experiments using kinase-dead CaMKII as substrate, short incubation times (30–60 s) and low temperature (4 °C) were used, which in turn also reduced inter-holoenzyme phosphorylation (*Hanson et al., 1994*; *Rich and Schulman, 1998*). Here, we split the enzyme and substrate kinases and could readily detect concentration-dependent inter-holoenzyme phosphorylation at 37 °C. We worked at minimal active kinase concentration and used 4 µM kinase-dead CaMKII, in order to approach the physiological CaMKII concentration, which is estimated to be 20–100 µM in dendritic spines (*Otmakhov and Lisman, 2012*). We showed that CaMKII$^{WT}$ can phosphorylate CaMKII$^{KD}$ subunits from distinct holoenzymes in a concentration-dependent manner in vitro. We also observed inter-holoenzyme phosphorylation in mammalian cell extracts and confirmed that CaMKII$^{KD}$ holoenzymes can associate with CaMKII$^{WT}$ holoenzymes under TIRF.

Evidence indicating that CaMKII spreads its kinase activity via subunit exchange between activated and unactivated holoenzymes came from single molecule fluorescence experiments, which monitored colocalization of differently labeled CaMKII under TIRF or by FRET (*Stratton et al., 2014*; *Bhattacharyya et al., 2016*). These studies identified activation-dependent colocalization of CaMKII subunits. We observed the same, but critically, if we triggered activation normally but then quenched it, before TIRF observation, the colocalization was entirely lost. At least two possibilities exist for obtaining this result. However improbable, subunits that exchanged during the activation period might have re-equilibrated to a low level of colocalization within their respective holoenzymes upon inhibition of the kinase reaction. Such selective reassembly seems far-fetched because our CaMKII$^{WT}$ subunits differed only by the identity of the fluorophore at the N-terminus. More likely, subunits did not exchange at all during the kinase reaction, but instead what we and others, *Stratton et al., 2014* have observed is the transient colocalization of holoenzymes during inter-holoenzyme phosphorylation.

The colocalization of kinase-dead and WT CaMKII subunits was weaker, which is expected if association requires kinase activity, because in this case only half of the sample was competent to perform phosphorylation. Importantly, this interpretation presents little conflict with existing data, which failed to detect colocalization of CaMKII$^{WT}$ and CaMKII$^{WT}$ incubated with the inhibitor bosutinib, used to mimic kinase-dead CaMKII (*Stratton et al., 2014*). According to our model, the absence of colocalization in CaMKII$^{WT}$ inhibited by bosutinib is in the nature of the inhibitor binding, which blocks the ATP-binding site (*Pellicena and Schulman, 2014*) and stabilizes compact inactive conformation of CaMKII (*Chao et al., 2011*). In such a conformation, CaMKII is not able to initiate trans-autophosphorylation, which manifests as a lack of colocalization, like in our TIRF experiments when ATP was omitted.

We directly probed subunit mixing with a photoactive crosslinker in the hub domain, but could not detect any mixing, nor did the restraint have any effect on kinase activity. The genetically encoded BzF crosslinker is monofunctional, and can capture unlabeled subunits at interfaces within ~5–10 Å (*Wittelsberger et al., 2008*). Even though crosslinking of the hub domain by BzF was not saturated, we observed bands of apparent high molecular weight above 0.3 MDa, yet the kinase activity (between untreated and crosslinked enzymes) was identical. In this context, we note that if subunit exchange were a prominent mechanism for spreading activity, then small molecules that stabilized the hub domain should reduce activity. However, binding of the putative CaMKII inhibitor GHB, which was shown to stabilize the hub domain, left substrate phosphorylation and trans-autophosphorylation untouched (*Leurs et al., 2021*). We could detect phosphorylation of CaMKII$^{KD}$ by CaMKII$^{F394BzF}$, but we could not detect kinase-dead protein in higher order oligomeric bands on western blot, which were exclusively from CaMKII$^{F394BzF}$. This result matches well with the much more promiscuous crosslinking of lysines by the bifunctional DSS crosslinker, which showed less crosslinking between subunit isotypes from different preparations (N14 vs N15), but abundant crosslinking within isotypes (N14:N14 and N15:N15).

How do holoenzymes interact during IHP? Crosslinks between heterotypic peptides detected by mass spectrometry were much more prevalent in activating conditions, reporting activity-dependent holoenzyme contacts with a spacing of 10 Å or less. We detected direct capture of substrate-active site peptide interactions (including phosphorylation of T286, *Figure 4—figure supplement 2* and *Supplementary file 1I & J*). Almost all peptide pairs were detected with an intensity ratio below one, meaning links between homo-isotypes (likely within holoenzymes) were more frequent than links between hetero-isotypes (between holoenzymes). This observation could in principle indicate that heterotypic holoenzymes (composed of mixed N14 and N15 subunits) were present, yet in the minority. However, the profile of crosslinks, being absent in all but the kinase domains for heterotypic links, speaks against the idea of subunit exchange. In particular, almost all crosslinks cover parts of the substrate binding cleft and the catalytic machinery. Heterotypic crosslinks were not detected in the hub domain. The mean ratio for kinase-kinase crosslinks was 25 ± 1% (32 crosslinks in activated conditions, 150 min incubation). If we assume that all heterotypic links come from inter-holoenzyme contacts, IHP might occur at about ¼ the rate of intra-holoenzyme autophosphorylation at 8 μM (in other words, with surprising speed; concentration-dependent phosphorylation was also estimated to be surprisingly fast by western blot see *Figure 1—figure supplement 3*). The flexible linkers might allow inter-holoenzyme contacts involving multiple kinase domains to act in a 'cooperative manner'. We illustrate this possibility in our speculative 'close mode' involving interdigitated holoenzymes, which reduces crosslink distances substantially (*Figure 5—figure supplement 1*). These observations further cast doubt on the rule that inter-holoenzyme autophosphorylation cannot occur. If a CaMKII dodecamer with mobile kinase domains were to abundantly phosphorylate its targets and auto-phosphorylate its own domains, but never auto-phosphorylate a neighbor despite being one of the most abundant cytosolic proteins, it should require as yet unknown additional factors which have been absent in all in vitro experiments to date.

Attempts to analyze the number of CaMKII molecules colocalized under TIRF by photobleaching of well-labeled CaMKII (*Theile et al., 2013*; *Elsner et al., 2019*) proved to be unreliable. The large number of subunits present means that shot noise overwhelms the stepwise fluorescence loss during single subunit bleaching (*Ulbrich and Isacoff, 2007*; *Chen et al., 2014*). Instead, we used mass photometry to obtain the distribution of molecular weights of CaMKII particles in basal and activating conditions. We detected activation-dependent appearance of particles corresponding to pairs of holoenzymes, supporting the idea that phosphorylation can occur in an inter-holoenzyme fashion (as well as the established intra-holoenzyme reaction). Similarly, self-association of CaMKII has previously been described in vitro (*Hudmon et al., 1996*; *Hudmon et al., 2001*), as well as in cells (*Hudmon et al., 2005*), where it was linked to ischemia-like conditions of lower pH. CaMKII clustering has also been detected on negative-stain EM grids (*Buonarati et al., 2021*), albeit no link to activation was asserted. Although peptides derived from the regulatory segment of CaMKII can break isolated hubs to lower molecular weight subunits in electrospray mass-spectrometry, the effect was absent in solution (*Karandur et al., 2020*). Consistent with this, we observed only a very minor fraction of monomer-dimer particles in activating conditions. Keeping in mind that the affinity for holoenzyme assembly is in low nM range (*Torres-Ocampo et al., 2020*), an observation that we could replicate (*Figure 7*), and that in the packed environment in dendritic spines, the concentration of CaMKII is in μM range (*Otmakhov and Lisman, 2012*), it is likely that CaMKII forms clusters of holoenzymes during activation, where more than two holoenzymes participate, but we did not detect a long tail in the mass distribution corresponding to larger supercomplexes.

What are the advantages or disadvantages of inter-holoenzyme phosphorylation? First of all, we should consider the dodecameric form of CaMKII. To avoid runaway phosphorylation, inter-holoenzyme phosphorylation requires concurrent phosphatase activity – but this is expected to be provided by PP2A (*Colbran, 2004*). Under persistent phosphatase attack, dodecamers can rescue their fully-phosphorylated state and autonomous activity, through intra-holoenzyme trans-autophosphorylation. In contrast, isolated CaMKII subunits in lower order forms (which should exist in the subunit exchange model) would be vulnerable to terminal dephosphorylation, destroying the signal. Gains in reliability may be offset against speed or scope. In principle, subunit exchange in tandem with rapid intra-holoenzyme autophosphorylation could grow CaMKII activity exponentially, albeit with a slower initial rate, because of the time to break and reform dodecamers. Inter-holoenzyme phosphorylation should proceed with a faster initial rate but be less prone to saturate local stocks of holoenzymes. Indeed,

measurements in dendrites suggest a steady summation of CaMKII activity (*Chang et al., 2019*; *Tao et al., 2021*), not a runaway exponential wave. Capture of peptides including phosphorylated T286 were only about 8 x less abundant in heterotypes (see *Supplementary file 1I & J* and *Figure 5—figure supplement 1*). The crosslinks obtained from mass-spectrometry suggest that CaMKII kinase domains are highly dynamic. Therefore, the loss of reaction rate from a lack of proximity might be compensated by the larger degree of freedom for inter-holoenzyme kinase-kinase domain interactions. Cooperative phosphorylation, whereby contacts made during inter-holoenzyme phosphorylation may be augmented by additional interactions, is naturally facilitated by the dodecameric ring architecture. Fixed dodecameric holoenzymes should have a larger fraction of their time to perform downstream signaling, unlike holoenzymes that must first disassemble and reassemble to spread activity. In our model of inter-holoenzyme phosphorylation, active dodecamers are a finished product, whose activity depends only on the rate constants of phosphorylation and stochastic attack of phosphatases. On the basis of this model, we speculate that there should be some dependence of phosphorylation spread on linker length, although we have not measured this. Further work will inform, for example, whether isoforms with short linkers propagate activity more slowly due to steric restrictions.

## Materials and methods

### Expression constructs

The DNA sequence encoding human *wild-type* CaMKII lacking the first 5 residues, and with a 30-residue linker (CaMKII[WT]), was codon optimized for expression in *E. coli*, and synthesized (GenScript) in the pET28b+plasmid, with flanking restriction sites NcoI and HindIII. Mutations K42R (forward: 5'-GAATACGCGGCGAGGATCATTAAC-3'; reverse: 5'-GTTAATGATCCTCGCCGCGTATTC-3'), D135N (forward: 5'-GGTTCACCGTAATCTGAAACCG-3'; reverse: 5'-CGGTTTCAGATTACGGTGAA CC-3'), F394TAG (forward: 5'-GGTCTGGATTAGCACCGTTTCT-3'; reverse: 5'-AGAAACGGTGCT AATCCAGACC–3'), and H418TAG (forward: 5'- CCTGAACCCGTAGATTCACCTGATG –3'; reverse: 5'- CATCAGGTGAATCTACGGGTTCAGG – 3') were introduced by site-directed mutagenesis. The AviTag was added to the C-terminus of CaMKII[WT] and CaMKII[K42RD135N] by splice overlap extension PCR, using forward AviTag primer: 5'- TTTCTGCGCTTCAAAAATATCGTTCAGGCCGGACCCGTGC GGCAGAACGCTCG-3' and reverse AviTag primer 5'- ATTTTTGAAGCGCAGAAAATTGAATGGC ATGAATAAAAGCTTGCGGCCGCACTC – 3', combined with vector specific forward: 5'-CGGCCGCA AGCTTTTATTCATG-3' and reverse: 5'-GATATACCATGGGCCACCAC-3' primers. In order to express CaMKII in mammalian cells, human CaMKII[WT], and human CaMKII[K42RD135N] with a C-terminal AviTag were cloned into pEGFP-N1 (Clontech) vector, with an IRES sequence inserted between CaMKII gene and EGFP. Rat CaMKII[WT] with an N-terminal EGFP pCAG-mEGFP-CaMKIIa was a gift from Ryohei Yasuda (Addgene plasmid # 127389; http://n2t.net/addgene:127389; RRID:Addgene_127389; *Chang et al., 2019*). The sortase A pentamutant (eSrtA) in pET29 was a gift from David Liu (Addgene plasmid # 75144; http://n2t.net/addgene:75144; RRID:Addgene_75144, *Chen et al., 2011*) and pET21a-BirA (Biotin Ligase) was a gift from Alice Ting (Addgene plasmid # 20857; http://n2t.net/addgene:20857; RRID:Addgene_20857, *Howarth et al., 2005*). The vector encoding Lambda Phosphatase (LPP) was a gift from John Chodera, Nicholas Levinson & Markus Seeliger (Addgene plasmid # 79748; http://n2t.net/addgene:79748; RRID:Addgene_79748; *Albanese et al., 2018*). His-tagged Syntide 2 (PLARTLSVAGLPGKK) - GST was cloned into the pETM11 vector.

### Protein expression

All CaMKII constructs (Kanamycin resistance) were co-expressed with Lambda protein phosphatase (LPP, Spectinomycin resistance) (*Chao et al., 2011*) in *E. coli* BL21 DE3 cells (NEB, catalog # C2527H), in LB medium, supplemented with 1 mM $MnCl_2$, which is a co-factor of LPP. Cultures were grown at 37 °C, until $OD_{600}$=0.8–1, and then protein expression was induced by adding 0.4 mM IPTG. Expression was continued overnight at 20 °C.

For expression of $^{15}N$-incorporated CaMKII, we co-transformed BL21 *E. coli* with CaMKII[WT] and LPP and grew them at 37 °C in minimal media (2 mM EDTA, 3.5 mM $FeSO_4$, 0.4 mM $ZnCl_2$, 0.06 mM $CuSO_4$, 1 mM $MgSO_4$, 0.3 mM $CaCl_2$, 50 mM $Na_2HPO_4$, 25 mM $KH_2PO_4$, 10 mM NaCl, 0.5% glucose, 1.5 μg/mL Thiamine, 0.15 μg/mL Biotin) supplemented with 0.75 mg/mL of $^{15}NH_4Cl$ as a source of $^{15}N$

isotope. Expression was induced by adding 0.5 mM IPTG at $OD_{600}$=0.8–1, and the expression was continued overnight at 20 °C shaking at at200 rpm.

Expression of BzF constructs was enabled by co-transformation of BL21 cells with individual plasmids containing a CaMKII amber (TAG) mutant, LPP, and orthogonal aminoacyl synthetase (aaRS) and tRNA from *M. jannaschii* (*Young et al., 2010*), which was a kind gift from Prof. Dr. Thomas Söllner. aaRS and tRNA were in a pEVOL vector under control of the araBAD promoter. In order for CaMKII to incorporate BzF at a desired position, the growth media was supplemented with 1 mM BzF at $OD_{600}$=0.6 at 37 °C. After 30 min, the pEVOL vector expression was induced by addition of 15 mM arabinose. Finally, at $OD_{600}$=1.2–1.5, CaMKII and LPP protein expression was induced by addition of 0.4 mM IPTG, and the media was supplemented with 1 mM $MnCl_2$. Expression was continued overnight at 20 °C, shaking at 220 rpm.

## Protein purification

Cell pellets were lysed in lysis buffer (50 mM Tris pH 8, 300 mM NaCl, 20 mM imidazole, 1 mM TCEP) supplemented with 0.02 mg/mL DNaseI, 0.5 mg/mL lysozyme and 1 mM PMSF. The lysates were additionally passed through an Avestin C5 cell disruptor twice, before the final centrifugation step (16 k rpm, 4 °C). All CaMKII constructs had an N-terminal $His_6$ tag, which was used for affinity purification with 5 mL NiNTA column (GE Healthcare), which was previously equilibrated in $Ni_A$ buffer (50 mM Tris pH8, 300 mM NaCl, 20 mM imidazole, 1 mM TCEP). The column was then extensively washed (100 mL of $Ni_A$ buffer, 50 mL of $Ni_A$ buffer with 50 mM imidazole, 50 mL of $Ni_A$ buffer with 80 mM imidazole), and the target protein was eluted with a gradient elution from 80 mM imidazole to 1 M imidazole (50 mM Tris pH8, 300 mM NaCl, 1 mM TCEP, 1 M imidazole) over 10 column volumes. Peak fractions were then pooled, imidazole concentration lowered to less than 100 mM with $Ni_A$ and the sample was incubated with 0.01 mg/mL final TEV (made in-house) in order to cleave off the $His_6$ tag. On the following day, the cleaved protein was concentrated to 500 µL, using 50 kDa cut off concentrator (4 mL Amicon R Ultra), and injected to Superose 6 10/300 column (GE Healthcare) previously equilibrated in size exclusion chromatography (SEC) buffer (25 mM Tris pH 8, 250 mM NaCl, 1% glycerol, 1 mM TCEP). Elution profile comprised of 2 peaks, one of which corresponded to molecular weight of dodecameric CaMKII (around 650 kDa) and the other one to 40 kDa CaMKII fragment, which is usually present during CaMKII purification from *E. coli*. The peak corresponding to dodecameric CaMKII was pooled and further concentrated to approximately 2 mg/mL. We obtained around 0.5–1 mg of protein per 2 L of *E. coli* pellet.

His tagged Syntide 2 – GST was purified from BL2-RIL cell pellets, using a two-step purification process. Frist, the cell pellet was resuspended in lysis buffer (50 mM Tris-HCl pH 7.5, 150 mM NaCl, 20 mM imidazole, 1 mM DTT) containing protease inhibitors (cOmplete, Roche) and further lysed by sonication. Upon centrifugation (16 k rpm, 4°), cleared cell lysate was loaded on a HisTrap Crude column (Cytiva) and eluted with a linear gradient of elution buffer (20 mM Tris-HCl pH 7.5, 300 mM NaCl, 500 mM imidazole, 1 mM DTT). Protein containing fractions were pooled, incubated with TEV protease, and dialysed against lysis buffer o/n. Digested sample was re-run on a HisTrap Crude column. The flow through was collected, concentrated and run on a High Load Superdex 75 26/60 size exclusion column (Cytiva), equilibrated in SEC buffer (20 mM PIPES pH 7.5, 50 mM NaCl). Protein containing fractions were pooled, concentrated to >20 mg/mL and flash frozen in liquid nitrogen.

Calmodulin from human was codon-optimized for expression in *E. coli* and purified according to published protocols (*Putkey and Waxham, 1996*).

BirA was expressed and purified according to the protocol from *Fairhead and Howarth, 2015*.

SrtA was expressed and purified according to *Popp, 2015*.

## Kinase assay with purified proteins

Kinase assays for western blot detection were performed at 37 °C, for 1 hr, with samples taken at 1-, 2-, 5-, 10-, 15-, 30-, and 60-min time points. Each time point was done in triplicate. Each reaction was performed in 10 µL volume with final concentrations as follows: 10 nM CaMKII[WT], 4 µM CaMKII[K42RD135N]-AviTag, 2 µM CaM (or 100 nM in 'low CaM condition'), 2 mM $CaCl_2$, 10 mM $MgCl_2$, 125 mM NaCl, 25 mM Tris pH 8, 2 mM TCEP, 100 µM ATP. First, 5 µL of 2 x master mix containing 20 nM CaMKII[WT], 8 µM CaMKII[K42RD135N]-AviTag and 4 µM CaM, in SEC buffer was distributed in PCR tubes. Then 2 µL of 5 x reaction buffer (10 mM $CaCl_2$ and 50 mM $MgCl_2$) was added to each tube. Finally, 3 µL of 350 µM

ATP was added to each tube, using a multi-channel pipette, starting from the longest time point (60 min). In order to quench the reactions, 10 µL of 4 x standard sample loading buffer was added using a multichannel pipette, starting with the shortest time point (1 min), followed by heat denaturation (2 min at 95 °C). Quenched reactions were kept at 4 °C before a western blot was run on the following day. Five µL of each quenched reaction was loaded on 12% SDS-PAGE gel, giving total of 1–2 ng of CaMKII$^{WT}$ and around 500 ng of CaMKII$^{K42RD135N}$-AviTag per lane.

The kinase assay for radioactivity detection was performed like described above, with several modifications. Briefly, the final reaction volume was 20 µL with final concentrations as follows: 15 nM CaMKII$^{WT}$, 4.4 µM CaMKII$^{K42RD135N}$-AviTag, 2 µM CaM (or 100 nM in "low CaM" condition), 2 mM CaCl$_2$, 10 mM MgCl$_2$, 0.1% BSA, 125 mM NaCl, 25 mM Tris pH 8, 2 mM TCEP, 100 µM ATP. Each time point was performed in triplicate. First 11 µL of master mix was distributed in PCR tubes, followed by the addition of 4 µL of 5 x reaction buffer (250 mM PIPES pH 7.2, 10 mM CaCl$_2$ and 50 mM MgCl$_2$, 0.5% BSA), and finally 5 µL of ATP. ATP stock was made by adding 2 µL of P$^{32}$ radioactive ATP to 400 µM stock of regular P$^{31}$ ATP. Reactions were then sampled like described above, and stopped by adding 10 µL of 4 x standard sample loading buffer. 10 µL of each sample was loaded on 12% SDS-PAGE gel, giving total of 5 ng of CaMKII$^{WT}$ and around 1.5 µg of CaMKII$^{K42RD135N}$-AviTag per lane. To probe substrate phosphorylation of CaMKII$^{F394BzF}$, instead of CaMKII$^{K42RD135N}$-AviTag, we used 40 µM of GST-Syntide-2, and sampled the kinase reactions as described above.

The kinase reactions done with different concentrations of CaMKII$^{WT}$ were performed in the same way, with the exception of changing CaMKII$^{WT}$ concentration (0.5 nM, 2 nM, 10 nM and 100 nM). The final concentration of Calmodulin in these reactions was set to 2 µM, and CaMKII$^{K42RD135N}$-AviTag to 4 µM. The reactions were performed at 37 °C in PCR tubes. Each time point (1, 2, 5, 10, 15, 30 and 60 min) was performed in triplicate and phosphorylation on T286 determined with western blotting. The ability of CaMKII$^{K42RD135N}$-AviTag to autophosphorylate was assessed in parallel, by incubating CaMKII$^{K42RD135N}$-AviTag with Ca$^{2+}$:CaM and ATP:Mg$^{2+}$ for 1 hr at 37 °C, without CaMKII$^{WT}$. The signal coming from autophosphorylation of CaMKII$^{WT}$ in these reactions was negligible for all except 100 nM, as assessed by incubation of CaMKII$^{WT}$ (at either 0.5 nM, 2 nM, 10 nM or 100 nM) with Ca$^{2+}$:CaM and ATP:Mg$^{2+}$ but without CaMKII$^{K42RD135N}$-AviTag, for 1 hr at 37 °C. At 100 nM, the signal from CaMKII$^{WT}$ remained about 20% of that from the CaMKII$^{K42RD135N}$-AviTag signal after 1 h (*Figure 1—figure supplement 3D*). All controls were performed in triplicate. The volume of reactions loaded on the gels was the same like the volume of reactions which included CaMKII$^{K42RD135N}$-AviTag.

Another set of controls, using similar reaction conditions, was performed for the kinase-dead CaMKII. Final concentration of CaMKII$^{K42RD135N}$-AviTag in 10 µL reactions was set to 4 µM, and CaMKII$^{WT}$ to 10 nM (or the WT protein was left out, where indicated in the figure). The reactions were performed in the presence or absence of Ca$^{2+}$:CaM (2 µM) and in the presence or absence of ATP:Mg$^{2+}$ (100 µM) for 30 min at 37 °C. The phosphorylation on T286 of CaMKII$^{K42RD135N}$-AviTag was determined by western blotting.

## HEK cell transfection and lysis

HEK 293 cells (ACC-305) were purchased from Leibniz Institute DSMZ (German Collection of Microorganisms and Cell Cultures GmbH, Braunschweig). According to the DSMZ, STR analysis according to the global standard ANSI/ATCC ASN-0002.1–2021 (2021) resulted in an authentic STR profile of the reference STR database. In-house testing for mycoplasma yielded a negative result.

A total of 1.5 million cells were seeded per well of a 6-well dish, and on the following day the cells were transfected using Polyethylenimine (PEI). For expression of single constructs, 1 µg of plasmid DNA was used with 3 µL of PEI, and mixed in 250 µL of Opti-MEM, vortexed briefly, left for 20–30 min for complexes to form and finally added to respective wells in a drop-by-drop manner. For co-expression, 1 µg of each construct was mixed in the same microcentrifuge tube with 6 µL of PEI in 250 µL of Opti-MEM. The medium was exchanged after 5 hr to fresh 6% FBS DMEM medium, and cells were left expressing the constructs for 48 hr before harvesting and cell lysis. In order to harvest the cells, the medium was first aspirated, and the cells were resuspended in 1 mL of ice cold DPBS per one well, and placed in ice cold microcentrifuge tubes, on ice, until a centrifugation step. After centrifugation at 2000 rpm for 10 min, the cells were snap frozen in liquid nitrogen and stored at –80 °C until further use. Cell lysis was performed in 100 µL of lysis buffer (per well): 50 mM Tris pH 8, 150 mM NaCl, 1% Triton X 100, 1 mM TCEP, supplemented with protease inhibitors (1 µM Pepstatin A, 10 µM

Leupeptin, 0.3 µM Aprotinin, 1 mM PMSF), 5 mM NaF, 0.02 mg/mL DNaseI, and 5 mM MgSO$_4$. After resuspension, the cells were left on ice for 20 min, with brief vortexing every 5 min. The lysates were then cleared by high-speed centrifugation for 15 min at 4 °C in a table-top centrifuge. Supernatants, containing CaMKII, were taken for the kinase assay.

### Kinase assay using HEK cell lysates with overexpressed CaMKII

A volume of 20 µL of each cleared cell lysate (CCL) was combined with 22 µL of lysis buffer and 12 µL of 4 x SDS-PAGE loading dye and heated at 95 °C for 2 min. Of this mixture, 20 µL was then loaded per well of 3–8% SDS-PAGE gel, and further subjected to western blotting in order to assess activity of overexpressed CaMKII.

Kinase assay was performed by incubating 20 µL of CCL with either 20 µL of lysis buffer or 20 µL of another CCL, under activating conditions with extra added 0.5 µM purified CaM and 500 µM ATP, at different temperatures (4°C and 37°C) and different incubation times (1 min or 10 min). Additionally, 20 µL of CCL containing overexpressed CaMKII$^{WT}$ was incubated with 8 µM purified CaMKII$^{K42RD135N}$-AviTag, under activating conditions. The reaction was quenched by adding 4 x SDS-PAGE loading dye, and heating at 95 °C for 2 min.

### Western blot detection and analysis

Samples were run on a 3–8% pre-cast gels (Thermo Scientific), in 1x TA buffer (Thermo Scientific) on ice for 1 hr and 10 min at 180 V. Transfer to pre-activated PVDF membrane (Millipore) was performed for 2 hr at 50 V at 8 °C, using Criterion Blotter (BioRad) in Towbin's transfer buffer with 20% MeOH. Membranes were blocked in Intercept Blocking Buffer (IBB) (LiCor) for 1 h, shaking at room temperature. Incubation with primary antibodies was done overnight at 8 °C. Membranes were incubated at the same time with a mixture of primary antibodies. Rabbit anti-pT286 primary antibody (CellSignalling Technologies, product #12716) was diluted 1:1000, while mouse anti-AviTag antibody (antibodiesonline, clone number 1D11D10) was diluted 1:400 in IBB, supplemented with 0.01% tween and 0.02% Na$_2$N$_3$. Anti-pan CaMKII antibody was also used (instead of anti-AviTag antibody) at 1:1000 dilution in IBB, supplemented with 0.01% tween and 0.02% Na$_2$N$_3$ (CellSignalling Technologies, product #50049S). On the following day, membranes were washed 3 times for 10 min in TBS-T buffer, and incubated with fluorescently-labeled secondary antibodies (IRDye 800CW Goat-anti Rabbit IgG, catalog # P/N 926–32211 and IRDye 680LT Goat-anti Mouse IgG, catalog # P/N 926–68020 from LiCor) for 1 hr at room temperature, shaking. A mixture of two secondary antibodies was diluted 1:10 000 in IBB buffer, supplemented with 0.01% tween and 0.01% SDS. The two secondary antibodies, goat-anti rabbit and goat-anti mouse, are excited at 800 nm and 700 nm, respectively. After secondary antibody incubation step, membranes were washed 3 times for 10 min in TBS-T and imaged using the LiCor imager. Blots were quantified using ImageJ/FIJI, and measured densitometry was plotted using IgorPro 9 (Wavemetrics, Lake Oswego, OR).

We normalized the pT286 signal to the total CaMKII signal at each time point. For simple measurements of phosphorylation time course, the data were fit using a Langmuir function. At concentrations of CaMKII$^{WT}$ below 10 nM, the reaction clearly has two components. Therefore, for measurements of concentration-dependent phosphorylation, we fitted each individual set of reactions with either one or two component exponential decays to obtain an estimate of the maximum. We then combined the data sets (each normalized to the maximum of its fitted curve) and fitted two component exponential decays to the data (weighted according to the SD) using IgorPro 9. We did fits using the same two component function for each curve:

$$f(t) = 1 - A_1 * \exp(-r_1 * t)A_2 * \exp(-c * k_2 * t)$$

$A_2$ was fixed to be $(1 - A_1)$. $c$ was the concentration of CaMKII$^{WT}$. At 100 nM, the reaction saturated and we estimated only a single component by fixing $A_1$ to zero. $r_1$ represents a phosphorylation rate that is concentration independent and $k_2$ represents a phosphorylation reaction dominated by concentration dependence (such as IHP).

### Radioactivity detection

Radioactive kinase reactions were run on 12% SDS-PAGE gels, followed by gel fixation and drying. The radioactivity was visualized using a photostimulable phosphor plate. Gel images were quantified

using ImageJ/FIJI. Measured densitometry was plotted using IgorPro 9, and fitted with a Langmuir isotherm.

## UV-induced crosslinking

In order to induce crosslinking of BzF to neighboring residues in CaMKII[BzF] mutants, we used a UV LED which can emit UV light of 365 nm wavelength (Opsytec Dr. Gröbel). The crosslinking was performed in the cold room, with reactions taking place in milled indentations in a passively-cooled custom stainless steel plate, to avoid excessive heating of the sample. CaMKII concentration during UV illumination was 8 µM in final volume of 20–40 µL. Each sample was treated five times with fifteen 1 s pulses, with 1 s break between pulses. After the illumination, samples were mixed with appropriate amount of SDS-PAGE loading dye, heated at 95 °C for 2 min and loaded on 3–8% SDS-PAGE gel. Around 1–2 ug of total protein was loaded per lane. Gels were run on ice for 1 hr 10 min at 180 V, and later stained with Coomassie dye or subjected to western blotting.

## Protein labeling for TIRF imaging

Biotinylation of CaMKII-AviTag was performed for 2 hr on ice in 100 µL final reaction volume, following an established protocol (Stratton et al., 2014) with small modifications. The following final concentrations of components were used: 50 µM CaMKII-AviTag, 2.5 µM BirA, 2 mM ATP, 5 mM MgCl$_2$, and 150 µM biotin. After biotinylation, the protein was subjected to fluorescent labeling using SrtA, following already published protocols (Theile et al., 2013).

On the previous day, SrtA recognition peptide (CLPETGG) was fluorescently labeled by incubation with maleimide dyes (ATTO-Tech) in 1.3 x excess dye over peptide (4.8 mM peptide and 6.24 mM ATTO-488; 4.22 mM peptide and 5.25 mM ATTO-594) in the dark at room temperature. The peptide labeling reaction was quenched on the following day by adding 1 µL of concentrated BME, and 50 µL of labeling buffer (100 mM Tris, pH 7.5, 150 mM KCl) to the final reaction volume of 120 µL. The sortase labeling reaction was incubated for 3 hr at room temperature in 400 µL final volume. The reaction contained 100 µL of biotinylation reaction, 12.5 µM SrtA, 40 µL of 10XSrtA buffer (500 mM Tris pH 7.5, 1.5 M NaCl, 100 mM CaCl$_2$), 120 µL of labeled peptide, 120 µL of SEC buffer. In order to get rid of BirA and SrtA and excess labeled peptide, the sample was first incubated with cobalt beads (GE Healthcare) to remove BirA and SrtA which both featured an N-terminal His-tag. The protein-containing supernatant was then applied to a PD-10 Sephadex desalting column (GE Healthcare) to remove excess of peptide/dye, and eluted using SEC buffer. Elution fractions were analysed for the presence of labeled CaMKII by running SDS-PAGE and imaging the gel on a BioRad imager, using the wavelength appropriate to each ATTO dye. Fractions containing labelled CaMKII were pooled and concentrated using 50 kDa cut-off concentrator tube (4 mL Amicon R Ultra) to approximately 0.5 mg/mL in 150–200 µL. We measured absorbance of labeled proteins using NanoDrop at $A_{280}$ and either $A_{488}$ or $A_{594}$ as appropriate, using extinction coefficients for CaMKII-AviTag ($E_{280}$=72.3 M$^{-1}$cm$^{-1}$) and Atto dyes ($E_{488}$=90 M$^{-1}$cm$^{-1}$, $E_{594}$=120 M$^{-1}$cm$^{-1}$). We estimated labeling efficiency to be around 80–90% by this measure.

## Preparation of cover slips for TIRF imaging

To reduce background signal and sources of autofluorescence, the coverslips (High Precision 1.5 H, 25 mm diameter from Marienfeld GmbH) were cleaned thoroughly, following an established protocol (Paul and Myong, 2022). Briefly, the coverslips were first scrubbed with 5% alconox using gloved hands. Then, they were sonicated in a sonicator bath (Elmasonic S 30 H Sonicator from Elma) for 20–30 min in MilliQ water. Next, they were etched with 1 M KOH for 30–45 min. Then, they were rinsed with water and dipped in 100% EtOH before being passed through a butane flame 4–5 times and dried completely. The next step consisted in coating with aminosilane ((3-Aminopropyl)triethoxysilane from Sigma Aldrich) for 30–45 min. For this, 1 mL of aminosilane was diluted with 11 mL of acetone before being put on the coverslips. After rinsing with water, the coverslips were then coated with a mixture of 98% mPEG (5 kDa) and 2% biotin-PEG (Silane-PEG-biotin 5 kDa from abbexa) in 0.1 M sodium bicarbonate (pH 8.5). Two coverslips were incubated on top of each other, with the PEG-solution in between, overnight. The next day, the coverslips could be separated, rinsed, dried completely and then stored at –20 °C until use. Just before use, the coverslips were taken out of the freezer, left to equilibrate at room temperature for 30 min, and then coated with NeutrAvidin, by

incubating each slide with 2 mL of 0.1 mg/mL Neutravidin dissolved in PBS (*Stratton et al., 2014*). The coverslips were then washed 10 times with 1 mL PBS prior to sample addition and imaging.

## Preparation of samples for TIRF imaging

We added 2 µL of labelled protein (8 µM) to 300 µL of imaging buffer (25 mM Tris pH 8, 150 mM KCl, 1 mM TCEP). If needed for the experiment, 150 nM of CaM/$Ca^{2+}$, 1 mM ATP and 2 mM $MgCl_2$ were added and incubated for 2 min at 37 °C. The sample was then added on the coverslip, incubated for 1 min and then rinsed three times with imaging buffer. Coverslips were kept in 1 mL of imaging buffer during imaging. In order to chelate $Ca^{2+}$ from CaM, we first incubated $Ca^{2+}$:CaM (where $Ca^{2+}$ is at 5 mM) with 15 mM BAPTA for 5 min at room temperature, and then added this mixture to the rest of the reaction. Final concentration of BAPTA was 0.25 mM, prior to putting the sample on the coverslip and washing.

For chelation of $Mg^{2+}$ ions, we first incubated the kinase reaction as usual (see above), and after 5 min we added 10 mM EDTA (to chelate 2 mM $Mg^{2+}$), incubated for 5 min at room temperature, and then added the sample to the coverslip, washed 3 times with imaging buffer and imaged.

## TIRF imaging

Imaging was performed using a Nikon Ti2 Eclipse microscope with a temperature-controlled stage (set to 37 °C). Excitation light was provided by alternating illumination with 488 nm (diode) and 561 nm (DPSS) lasers housed in an Omicron Lightbox. The laser light was filtered using a quad band notch filter (400-410, 488, 561, 631-640) and reflected through the 60 x Apo TIRF 1.49 NA objective by a quad band dichroic filter (zt405/488/561/640 rpc; from set F73-400). Coverslips were illuminated in TIRF mode and the angle was adjusted in software (Nikon NIS elements) as was the motorised stage and autofocus system. Emission light was filtered with 525/40 (488 nm excitation) or 600/52 filters (561 nm excitation, both Semrock). Images were recorded with a Prime95B 22 mm sCMOS camera.

## TIRF image analysis

Colocalization analysis was done with ImageJ/ FIJI (*Schindelin et al., 2012*) and the colocalization Plug-In "JaCoP" (*Bolte and Cordelières, 2006*). To remove background, we set the threshold to the top 1% of signal and used it to create an image mask for each channel. These masks were then applied to the original image, to generate thresholded images used for colocalization analysis. In order to quantify the colocalization, the thresholded images were then analysed using 'JaCoP' Plug-In in FIJI and the Pearson's correlation coefficient was calculated. The Pearson's correlation coefficient is a measure of linear correlation between two sets of data, the value ranging from –1 for negative correlation to 1 for positive correlation. For each condition, five to six data sets were analyzed, from recordings made on multiple days. The significance was calculated using a multi-comparison test (Dunnett's test) with $\alpha$=0.05 in IgorPro 9.

## Mass photometry

Mass Photometry was performed on Refeyn One$^{MP}$ mass photometer. The unactivated samples of CaMKII were prepared by diluting 8 µM CaMKII$^{WT}$ in TIRF imaging buffer (25 mM Tris pH 8, 150 mM KCl, 1 mM TCEP) to obtain three different CaMKII stock concentrations – 1.6 µM, 0.4 µM, and 0.04 µM. The recordings were made by diluting 5 µL of each stock sample in 15 µL of imaging buffer, directly on the cover slide. The final concentration of CaMKII$^{WT}$ in the drop was 400 nM, 100 nM and 10 nM, respectively. Each dilution was measured three times. Activated sample was obtained by incubating 8 µM CaMKII$^{WT}$, 1.5 µM $Ca^{2+}$:CaM, 1 mM ATP, 2 mM $MgCl_2$ in imaging buffer. The sample was left incubating for 5 min at room temperature before stock dilutions were made with 1.6 µM, 0.4 µM and 0.04 µM final CaMKII$^{WT}$ by diluting the original incubation reaction in imaging buffer. Each sample dilution was measured three times, by diluting 5 µL of stock sample in 15 µL imaging buffer. The same procedure was repeated for CaMKII$^{K42R/D135N}$-AviTag.

Percentages of monomers participating in each peak were calculated in the following way: first peak area percentages were calculated for each peak. Then, the number of monomers for each peak was calculated with respect to the area percentage. Finally, the percentage of monomers in each peak was calculated based on total particles identified in each measurement. We used IgorPro9 multipeak fitting function to fit the peaks obtained by the instrument. Here is an example of the calculation used

for **Figure 7B**: Peak area % (calculated by the fitting software) for each peak is 4.7% for the 2-mer, 85.9% for the 12-mer, and 9.4% for the 24-mer. These are percentages of particles detected, not monomers. To calculate in terms of monomers, we multiplied each peak area % with the number of subunits and got 4.7x2 = 9.4 for 2-mer, 85.9x12 = 1030.8 for 12-mer and 9.4x24 = 225.6 for 24-mer. We summed these to obtain the total monomer number (1265.8) detected in this measurement. We obtained the fraction of monomers in each of the three peaks with respect to total number of particles, corresponding in this case to 0.7%, 81.5% and 17.8% of the monomers in the sample, respectively.

## Preparation of samples for mass spectrometry

First, proteins were exchanged into HEPES containing buffer (25 mM HEPES pH 8, 250 mM NaCl, 1% glycerol, 1 mM TCEP) using PD-10 Sephadex desalting column (GE Healthcare). The following samples (50 µL final reaction volume) were prepared for the kinase reaction: 8 µM $^{14}$N CaMKII$^{WT}$ unactivated, 8 µM $^{15}$N CaMKII$^{WT}$ unactivated, 4 µM $^{14}$N CaMKII$^{WT}$ and 4 µM $^{15}$N CaMKII$^{WT}$ unactivated, 8 µM $^{14}$N CaMKII$^{WT}$ activated (with 1.5 µM Ca$^{2+}$:CaM, 100 µM ATP, 200 µM MgCl$_2$), 8 µM $^{15}$N CaMKII$^{WT}$ activated (with 1.5 µM Ca$^{2+}$:CaM, 100 µM ATP, 200 µM MgCl$_2$), 4 µM N$^{14}$ CaMKII$^{WT}$ and 4 µM $^{15}$N CaMKII$^{WT}$ activated (with 1.5 µM Ca$^{2+}$:CaM, 100 µM ATP, 200 µM MgCl$_2$). The samples were left for 30 min or for 150 min at 37 °C. The samples were then incubated at room temperature with 2 mM DSS final concentration for 30 min, and the reaction was quenched with 30 mM Tris pH 8 for 15 min at room temperature. The samples were stored at –80 °C until further use.

## Sample digestion for mass spectrometry

Samples were denatured with 8 M Urea and reduced with 5 mM dithiothreitol (DTT) at 37 °C for 45 min, followed by alkylation with 40 mM chloroacetamide (CAA) for 45 min in the dark. The urea was then diluted to 1 M with 50 mM Triethylammonium bicarbonate buffer (TEAB), pH 8.0. Protein digestion was finished by using Lysyl endopeptidase C (Wako) with an enzyme-to-protein ratio 1:100 (w/w) for 3 h at 37 °C and continued with trypsin (Serva) at an enzyme-to-protein ration 1:50 (w/w) for overnight at 37 °C. Enzymically-digested samples were cleaned with C8 Sep-Pak (Waters), dried and stored at –20 °C for further analysis.

## LC-MS analysis

The LC-MS analysis of crosslinked samples were conducted by Orbitrap Fusion Lumos Tribrid Mass Spectrometer (Thermo Fisher Scientific) coupled with UltiMate 3000 RSLC nano LC system (Thermo Fisher Scientific). The separation of samples was performed on a 50 cm reverse-phase column (in-house packed with Poroshell 120 EC-C18, 2.7 µm, Agilent Technologies) with 180 min gradient. High field asymmetric waveform ion mobility spectrometry (FAIMS) was enabled with internal stepping −40/−50/−60 V. Cross-linked samples were acquired with 120,000 resolution at MS1 levels, 30,000 resolution at MS2 levels, charge state 4–8 enabled for MS2, higher-energy collisional dissociation (HCD) at 30% for MS2 fragmentation.

## Data analysis

All raw spectra were searched with pLink2 2.3.9 (**Chen et al., 2019**), against the protein sequence of CaMKII (Uniprot ID: Q9UQM7). Only $^{14}$N labeled crosslinks were searched. pLink parameters were as follows: minimum peptide length, 6; maximal peptide length, 60; missed cleavages, 3; Cys carbamidomethyl (57.021 Da) as fixed modification and Met oxidation (15.995 Da) as variable modification; LinkerMass was set to 138.068 Da and MonoMass was set to 156.079 Da; 10 ppm for precursor mass tolerance and 20 ppm for fragment mass tolerance; 1% separate FDR at CSM level. Raw files and search result were imported into in-house developed R-script using the rawrr package (**Kockmann, 2021**) to extract the MS1 intensities of uni-isotope and mixed-isotope crosslinks. For single spectrum of identified cross-linked peptides, the intensity ratio for crosslinks was evaluated with the following equation:

$$R = \frac{I^{14}N^{15}N + I^{15}N^{14}N}{I^{14}N^{14}N + I^{14}N^{15}N}$$

where *I* was the highest centroid peak intensity in centroid envelope from the MS1 mass spectrum.

## Structural analysis of crosslinks

Structures were visualized in ChimeraX (1.4, *Pettersen et al., 2021*). The XMAS plugin was used for a preliminary analysis. Subsequently, crosslinks were imported as pseudobonds between C-alpha atoms. The pseudobond files were generated from mass spectra pLink files in Excel. To generate illustrative holoenzyme interactions and 'close mode', the holoenzymes were moved into place by hand, taking care to minimize close residue clashes.

## Acknowledgements

We thank Marcus Wietstruk for technical assistance, Martin Lehmann (FMP Imaging facility) for support with TIRF imaging, the Söllner Group (University of Heidelberg) for the plasmids encoding BzF synthetase and tRNA in *E. coli* and for advice on in vitro LED crosslinking, Sascha Lange (FMP Solid State NMR) for help with expression and purification of $^{15}$N-labelled CaMKII and Heike Stephanowitz (FMP Structural Interactomics) for intact mass determination. IL was recipient of a Marie Curie Incoming International Fellowship (798696) and "Wiedereinstiegsstipendium" from the Leibniz FMP. This work was funded by the DFG TRR 186 (Project A07, Projektnummer 272140445 to AJRP) and a DFG Heisenberg Professorship (to AJRP, Projektnummern 323514590 & 446182550). Molecular graphics and analyses performed with UCSF ChimeraX, developed by the Resource for Biocomputing, Visualization, and Informatics at the University of California, San Francisco, with support from National Institutes of Health R01-GM129325 and the Office of Cyber Infrastructure and Computational Biology, National Institute of Allergy and Infectious Diseases. The article processing charge was funded by the Deutsche Forschungsgemeinschaft (DFG, German Research Foundation) – 491192747 and the Open Access Publication Fund of Humboldt-Universität zu Berlin.

## Additional information

### Funding

| Funder | Grant reference number | Author |
|---|---|---|
| HORIZON EUROPE Marie Sklodowska-Curie Actions | 798696 | Iva Lučić |
| Deutsche Forschungsgemeinschaft | TRR186/A07; project no. 278001972 | Andrew JR Plested |
| Deutsche Forschungsgemeinschaft | 323514590 & 446182550 | Andrew JR Plested |
| Boehringer Ingelheim | PhD Fellowship | Andreas Franz |
| Deutsche Forschungsgemeinschaft | TRR186/A15; project no. 278001972 | Markus C Wahl Florian Heyd |
| European Research Council | ERC-STG no. 949184 | Fan Liu |

The funders had no role in study design, data collection and interpretation, or the decision to submit the work for publication.

### Author contributions

Iva Lučić, Conceptualization, Data curation, Formal analysis, Supervision, Funding acquisition, Investigation, Visualization, Writing – original draft, Writing – review and editing; Léonie Héluin, Data curation, Formal analysis, Visualization, Writing – review and editing; Pin-Lian Jiang, Data curation, Formal analysis, Methodology, Writing – review and editing; Alejandro G Castro Scalise, Data curation, Formal analysis; Cong Wang, Data curation, Software, Formal analysis; Andreas Franz, Data curation, Visualization, Methodology, Writing – review and editing; Florian Heyd, Markus C Wahl, Funding acquisition, Writing – review and editing; Fan Liu, Resources, Software, Formal analysis, Funding acquisition, Methodology, Writing – review and editing; Andrew JR Plested, Conceptualization, Resources, Formal analysis, Supervision, Funding acquisition, Writing – original draft, Writing – review and editing

## Author ORCIDs

Iva Lučić https://orcid.org/0000-0002-8077-2195
Florian Heyd http://orcid.org/0000-0001-9377-9882
Markus C Wahl https://orcid.org/0000-0002-2811-5307
Andrew JR Plested https://orcid.org/0000-0001-6062-0832

## Decision letter and Author response

Decision letter https://doi.org/10.7554/eLife.86090.sa1
Author response https://doi.org/10.7554/eLife.86090.sa2

---

## Additional files

### Supplementary files

• Supplementary file 1. Crosslinked peptides identified by MS X-linking. (A) Homotypic crosslinks (basal, 30 min). (B) Homotypic crosslinks (basal, 150 min). (C) Heterotypic crosslinks (basal, 30 min). (D) Heterotypic crosslinks (basal, 150 min). (E) Homotypic crosslinks (activated, 30 min). (F) Homotypic crosslinks (activated, 150 min). (G) Heterotypic crosslinks (activated, 30 min). (H) Heterotypic crosslinks (activated, 150 min) (I) pT286 Heterotypic peptides (30 min). (J) pT286 Heterotypic peptides (150 min).

• MDAR checklist

### Data availability

All data generated during this study are included in the manuscript, supporting figures and supporting tables.

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
