## [Editor Report]

This manuscript reports the fundamental finding that an oligomeric protein kinase, CaMKII, can be phosphorylated by another molecule of the holoenzyme in a manner that does not involve subunit exchange. The evidence for the main conclusion is compelling, and supported by several independent experiments. If independently confirmed in the future, the study will stand as having provided a novel regulatory mechanism for the autophosphorylation of this kinase. The work will be of broad interest to molecular and cellular neuroscientists as well as biochemists.

---

## [Decision Letter]

**Decision letter after peer review:**

Thank you for submitting your article "CaMKII activity spreads by inter-holoenzyme phosphorylation" for consideration by *eLife*. Your article has been reviewed by 3 peer reviewers, and the evaluation has been overseen by a Reviewing Editor and Volker Dötsch as the Senior Editor. The following individuals involved in the review of your submission have agreed to reveal their identity: Karl Ulrich Bayer (Reviewer #1); M Neal Waxham (Reviewer #2); Margaret Stratton (Reviewer #3).

Essential revisions:

Two reviewers acknowledge that the manuscript provides novel insight into the mechanism by which CaMKII activity is transferred to the naïve molecule and that the findings are of broad interest to the neuroscience community. However, the discussion among all reviewers raises the following issues:

1) All reviewers agree that the title should be toned down. This is because the title makes claims that go beyond the assessment that trans-phosphorylation can occur between holoenzymes. The title suggested by the reviewers is "CaMKII autophosphorylation can occur between holoenzymes without subunit exchange". It is important to include "without subunit exchange" in the title because this will help all readers to find related papers through search engines. You may find an alternative from the reviewer's suggestions below.

2) Regarding subunit exchange, reviewers 1 and 2 believe that their conclusion is sufficiently supported by the current multi-pronged experimental evidence. Reviewer 3 would also concede that subunit exchange indeed does not occur under conditions in the manuscript but has some specific technical concerns that make some parts of the experimental evidence less compelling in her view.

3) All reviewers agree that it would be good to have the specific technical concerns and caveats listed in the public review. This should enable the reader to come up with a balanced evaluation of the strength of the evidence (and may help future investigations in the matter).

We would like you to respond to the reviewer's comments below.

*Reviewer #1 (Recommendations for the authors):*

This is Ulli Bayer, from the University of Colorado. I love the entire paper. Well, with the exception of the title. I would strongly suggest changing it. (Although this is of course entirely at the discretion of the authors). The reason why I love this paper is that it nicely and comprehensively debunks a myth about the propagation of pT286, i.e. the subunit exchange model. (I actually recently read the corresponding manuscript on bioRxiv and had meant to email to communicate as much, but then I was asked by *eLife* to provide a formal review). The reason why I don't like the current title is that it creates a new myth. Notably, to fix this sufficiently, not much else than the title would need to be changed.

*Reviewer #2 (Recommendations for the authors):*

A few clarifications and interpretative issues are suggested below to improve the paper.

One global point the authors should address is the idea of runaway IHP. The subunit exchange model is inherently self-limiting, where only replaced subunits would be subject to intraholoenzyme phosphorylation. This is not the case with IHP. Presumably, this could lead, perhaps quickly, to a situation where all holoenzymes within a local environment would exist in a phosphorylated state. On Page 14, lines 484-486, the authors take a step in this direction, but it doesn't confront the problem of potentially feeding forward autophosphorylation. Perhaps the authors are counting on the idea of an activating stimulus being necessary to drive IHP. If so, should be clearer about this thinking.

As an aside, no comment was made for the absence of CaM-bound forms of the kinase-dead holoenzyme (Figure S13). Some comment should be made on why this might be. Mass photometry is clearly sensitive enough, and presumably, the mutations used to create the kinase-dead mutant do not influence ca^2+^/CaM-binding. Are the CaM-binding properties of this double kinase-dead mutant known?

I could not identify in the legends how many times experiments were done, how many replicates within the experiment were completed, where are the error bars, etc. This should be made clearer in the legends for each figure. There is also shading in the line plots e.g., Figure 1D in blue, what is this meant to represent?

In Figure 1, Panel D the first time point already shows a 50% P/Pmax, why were reactions not terminated at shorter time points? It's hard to take the t_1/2_ seriously from this plot when the first data point looks to be at half-saturation.

On page 2, lines 57 and 58 – a note is made to the Lee et al. study, 2022. The statement is incorrect. The cited study does not study protein turnover it studies the approximate lifetime of CaMKII autophosphorylation.

On page 8, lines 254-256, the authors describe types of interactions from their cross-linking data that differ when compared to the Chao et al., linkerless CaMKII crystal structure, and the Myers et al., EM with linker CaMKII structure. I do not fully follow what is meant by interactions differ. Chao et al. described relatively tight interactions of the catalytic domain with the hub, but Myers et al. did not. The authors should make a more direct link to the comparison of their drawings for the reader.

On page 9, lines 292-294 the authors are describing the results of CaMKII holoenzyme interactions as being activity-dependent (agreed) and transient. First, no kinetics are done, so it might be good to constrain this terminology to the time frame of the experiment. But further, the lifetime of the CaMKII-CaMKII interaction is clearly long-enough to be captured by the process of diluting and binding the sample to the coverslips for subsequent analysis. While no specific claims are being made as to the lifetime, it must not be as transient as this statement might imply or the pairs of holoenzymes would have dissociated during preparation for imaging.

On page 13, lines 446-449, the authors comment on the unreliability of step photobleaching and relate that to shot noise preventing an accurate measurement. This comment is important for interpreting the data in the present manuscript in comparison to earlier studies using similar approaches to address the subunit exchange hypothesis (Stratton et al., 1996). This argument would be more convincing if the authors supply some of their photobleaching attempts in the SI that highlight how the measurements are compromised.

On page 20, lines 697-698, please provide the extinction coefficients used to make these calculations of labeling efficiency.

The title for Figure 2 legend – "Restricting CaMKII mobility…" does not seem an accurate statement. Perhaps more direct? Cross-linking CaMKII subunits does not….?

*Reviewer #3 (Recommendations for the authors):*

As stated in my public review, my major concerns are with UV crosslinking and TIRF microscopy. In my view, the results of these experiments are confounded by the way the experiments are set up and analyzed.

For the crosslinking – can the efficiency be increased? Or can the experimenters purify the 6mer crosslinked species and use this for their experiments? As it stands, Figure 2 and Figure 3 are not convincing due to low crosslinking.

The blue arrows indicating bands in Figure 3 are very unclear – these images need to be processed so that the reader can actually see these bands.

On a related note, since your crosslinking is incomplete, the argument for inter-holoenzyme phosphorylation is weakened. If the crosslinking is improved and you can show similar rates of transphosphorylation, this will be significantly stronger data.

For the single-molecule experiments, single particles (ROIs) should be selected based on threshold, shape, and size using the General Analysis module in Nikon software, since that's the type of microscope you're using. You should set minimum and maximum thresholds to ensure that you're analyzing a similar population of particles. If you're unable to select single particles for analysis, then this is not single molecule resolution and the experiment should be repeated at lower concentrations or on cleaner coverslides (whatever the issue may be). The images shown in the main paper appear not to be well-resolved, but some images in the supplement do look well-resolved, this just needs to be analyzed properly and ensure that the microscope is properly aligned. If you're able to analyze the intensities of single particles, this should be plotted as a histogram. According to your model, the histogram of particles under basal conditions should correspond to a single holoenzyme, under activation conditions, the intensities should correspond to a double holoenzyme (as you see a small population of these in MP), and after EGTA treatment the intensities should again correspond to a single holoenzyme. The authors claim 80-90% labeling efficiency but should explain how this is determined – because if this is true I would expect homogenous particles that are quite bright (12-14 fluorophores) under basal conditions.

---

## [Author Response]

Essential revisions:1) All reviewers agree that the title should be toned down. This is because the title makes claims that go beyond the assessment that trans-phosphorylation can occur between holoenzymes. The title suggested by the reviewers is "CaMKII autophosphorylation can occur between holoenzymes without subunit exchange". It is important to include "without subunit exchange" in the title because this will help all readers to find related papers through search engines. You may find an alternative from the reviewer's suggestions below.

The authors thank all reviewers for their advice, and will include the changed title into the revised manuscript.

2) Regarding subunit exchange, reviewers 1 and 2 believe that their conclusion is sufficiently supported by the current multi-pronged experimental evidence. Reviewer 3 would also concede that subunit exchange indeed does not occur under conditions in the manuscript but has some specific technical concerns that make some parts of the experimental evidence less compelling in her view.

We thank the reviewers for their assessments of our manuscript. We have now addressed some of the concerns raised by Reviewer 3.

3) All reviewers agree that it would be good to have the specific technical concerns and caveats listed in the public review. This should enable the reader to come up with a balanced evaluation of the strength of the evidence (and may help future investigations in the matter).

We also think this is a good idea. To the extent that it was feasible, we also tried to address the concerns in the manuscript.

Reviewer #1 (Recommendations for the authors):This is Ulli Bayer, from the University of Colorado. I love the entire paper. Well, with the exception of the title. I would strongly suggest changing it. (Although this is of course entirely at the discretion of the authors). The reason why I love this paper is that it nicely and comprehensively debunks a myth about the propagation of pT286, i.e. the subunit exchange model. (I actually recently read the corresponding manuscript on bioRxiv and had meant to email to communicate as much, but then I was asked by eLife to provide a formal review). The reason why I don't like the current title is that it creates a new myth. Notably, to fix this sufficiently, not much else than the title would need to be changed.

The authors thank Dr Bayer for his kind assessment of our paper. We agree that the paper would benefit from adjusting the title. We changed it accordingly.

Reviewer #2 (Recommendations for the authors):A few clarifications and interpretative issues are suggested below to improve the paper.One global point the authors should address is the idea of runaway IHP. The subunit exchange model is inherently self-limiting, where only replaced subunits would be subject to intraholoenzyme phosphorylation. This is not the case with IHP. Presumably, this could lead, perhaps quickly, to a situation where all holoenzymes within a local environment would exist in a phosphorylated state. On Page 14, lines 484-486, the authors take a step in this direction, but it doesn't confront the problem of potentially feeding forward autophosphorylation. Perhaps the authors are counting on the idea of an activating stimulus being necessary to drive IHP. If so, should be clearer about this thinking.

Kinase reactions performed in this study contained only CaMKII (wt or kinase-dead) with activation stimuli (Ca^2+^/CaM and Mg^2+^/ATP). No phosphatase was added. We assume that in the cell, phosphatases are also acting on CaMKII to dephosphorylate it (constantly) and in this way keep the brake on runaway IHP. On the other hand, processes controlled by CaMKII, like long-term potentiation, require long-term spread of the initial signal (Ca^2+^ signal). It is plausible to consider that spread of CaMKII phosphorylation, allowed by IHP, would be necessary for a more stable maintenance or amplification of the initial signal in the presence of phosphatases. We added some lines to this effect in the discussion line 505-507.

We will continue to explore the molecular details of this hypothesis in future work.

As an aside, no comment was made for the absence of CaM-bound forms of the kinase-dead holoenzyme (Figure S13). Some comment should be made on why this might be. Mass photometry is clearly sensitive enough, and presumably, the mutations used to create the kinase-dead mutant do not influence ca^2+^/CaM-binding. Are the CaM-binding properties of this double kinase-dead mutant known?

We did not directly test binding of Calmodulin to kinase-dead CaMKII, but since kinase-dead protein can become phosphorylated at T286 by WT CaMKII, we assume that it’s ability to bind Calmodulin is unaffected by the two mutations (K42R and D135N), which are far away from the Calmodulin binding site (residues 290-310). When Calmodulin is omitted from the phosphorylation reaction, the phosphorylation on CaMKII^KD^ does not occur (Figure 7—figure supplement 2C).

I could not identify in the legends how many times experiments were done, how many replicates within the experiment were completed, where are the error bars, etc. This should be made clearer in the legends for each figure. There is also shading in the line plots e.g., Figure 1D in blue, what is this meant to represent?

Thank you for this suggestion. The kinase assays were done in triplicate (technical replicates) in each experiments, but also as “biological” replicates (done with different protein preparations, on different days). The mixing experiments from Figure 3 were done twice (with kinase-dead and wild-type protein). The crosslinking mass spectrometry was done in duplicates (each condition was performed two times on different days, and analyzed two times). Mass photometry was done in triplicates (technical replicates) but also several times with different protein preparations on different days. We have now added the numbers of replicates in the legends of each Figure, as well as to the Materials and Methods part of the manuscript. The shading represents error bars. We are sorry for not being clear enough, and we now clarified this in the Figure legends.

In Figure 1, Panel D the first time point already shows a 50% P/Pmax, why were reactions not terminated at shorter time points? It's hard to take the t_1/2_ seriously from this plot when the first data point looks to be at half-saturation.

This is simply because of the technical nature of the experiment. Each time point was done in triplicate and conditions with high or low Calmodulin concentration were done at the same time for each time point (please see M and M section “Kinase assay with purified proteins”). The reaction was started in all tubes at the same time by adding ATP. The reactions were then stopped by adding STOP buffer at indicated time points (1, 2, 5, 10, 15, 30, 60 min). One minute is the shortest time point which could accommodate all these steps with reasonable accuracy. We could have tried 30 second, but shorter than that would be difficult to accommodate all the steps involved in starting and stopping the reaction, which include: putting the tips on the multichannel pipette, taking up ATP, dispensing ATP, shortly mixing by pipetting up and down a few times, discarding the tips, putting on new tips, taking up the STOP buffer, dispensing it and mixing it well for all 6 samples (triplicates of the same time point for 2 different Calmodulin concentrations) at the same time.

On page 2, lines 57 and 58 – a note is made to the Lee et al. study, 2022. The statement is incorrect. The cited study does not study protein turnover it studies the approximate lifetime of CaMKII autophosphorylation.

The reviewer is correct. The study indeed studies the approximate lifetime of CaMKII autophosphorylation, but one of the conclusions (indeed the title of the paper) is that the autophosphorylation outlives the known CaMKII protein turnover. The authors of this study measure CaMKII activity (using LTP as a read-out) to be present for 2 weeks in cultured brain slices, but the turnover of CaMKII molecules is 5-6 days. We modified the text around line 59 to acknowledge this distinction.

On page 8, lines 254-256, the authors describe types of interactions from their cross-linking data that differ when compared to the Chao et al., linkerless CaMKII crystal structure, and the Myers et al., EM with linker CaMKII structure. I do not fully follow what is meant by interactions differ. Chao et al. described relatively tight interactions of the catalytic domain with the hub, but Myers et al. did not. The authors should make a more direct link to the comparison of their drawings for the reader.

We meant to say that our crosslinking MS data shows a high degree of inter-domain interactions within the holoenzymes (kinase-kinase, kinase-hub, kinase-regulatory, kinase-linker, regulatorylinker, regulatory – hub, linker- linker, linker -hub, hub-hub), which are far more dynamic than those described in the negative-stain EM structure, where there is almost no interaction between the kinase domains and the hub, or regulatory domains with the hub. The same goes for the crystal structure in which the CaMKII holoenzyme is often described as a compact “rock”, while our crosslinking MS data show that the interactions within a holoenzyme are much more dynamic. The overall message of this passage should be that the holoenzyme is a highly dynamic structure, and we adjusted the text accordingly (please see the text around line 282).

On page 9, lines 292-294 the authors are describing the results of CaMKII holoenzyme interactions as being activity-dependent (agreed) and transient. First, no kinetics are done, so it might be good to constrain this terminology to the time frame of the experiment. But further, the lifetime of the CaMKII-CaMKII interaction is clearly long-enough to be captured by the process of diluting and binding the sample to the coverslips for subsequent analysis. While no specific claims are being made as to the lifetime, it must not be as transient as this statement might imply or the pairs of holoenzymes would have dissociated during preparation for imaging.

The term “transient” was used here to describe that the reaction is reversible by chelation of Mg^2+^ (so by stopping phosphorylation). We adjusted the text around line 330 to be clearer. We did not mean to imply any measurement of spontaneous dissociation of the holoenzymes after all CaMKII molecules have been phosphorylated.

On page 13, lines 446-449, the authors comment on the unreliability of step photobleaching and relate that to shot noise preventing an accurate measurement. This comment is important for interpreting the data in the present manuscript in comparison to earlier studies using similar approaches to address the subunit exchange hypothesis (Stratton et al., 1996). This argument would be more convincing if the authors supply some of their photobleaching attempts in the SI that highlight how the measurements are compromised.

We would like to work further on this kind of measurement before presenting any data. At the moment, we do see steps but cannot count them beyond a few. This is entirely predicted by previous work (subunit counting above about 5 is really difficult). The purpose of our comment is to make it clear that this is not a trivial measurement when 12 or 24 subunits are involved. We now gave two references to back to this up.

On page 20, lines 697-698, please provide the extinction coefficients used to make these calculations of labeling efficiency.

We have now included extinction coefficients at around line 758.

The title for Figure 2 legend – "Restricting CaMKII mobility…" does not seem an accurate statement. Perhaps more direct? Cross-linking CaMKII subunits does not….?

Thank you for this suggestion. We changed the title to “Crosslinking CaMKII subunits in the hub domain does not change the rates of trans-autophosphorylation”.

Reviewer #3 (Recommendations for the authors):As stated in my public review, my major concerns are with UV crosslinking and TIRF microscopy. In my view, the results of these experiments are confounded by the way the experiments are set up and analyzed.For the crosslinking – can the efficiency be increased? Or can the experimenters purify the 6mer crosslinked species and use this for their experiments? As it stands, Figure 2 and Figure 3 are not convincing due to low crosslinking.

We have designed and tested several BzF mutants in the hub domain of CaMKII. One of the criteria was that the BzF mutant elutes from the Size exclusion column as a dodecamer, assuring that the introduction of the BzF residue on its own did not compromise the interactions in the hub necessary to maintain the intact holoenzyme. We showed two of the best candidates in this paper (F394BzF and H418BzF), and neither of them showed slower phosphorylation kinetics upon crosslinking. We agree it would be great to purify the 6-mers, but this is not feasible in our hands, because these mutants form dodecamers in solution, and therefore always elute as 12-mers from the SEC column, regardless of UV exposure. If the subunits would mix during phosphorylation, as the prevailing mechanism for the spread of kinase activity (phosphorylation on T286 in kinase-dead protein) – we would expect to detect at least a portion of CaMKII^KD^-AviTag in our phosphorylation mixtures, as the kinase-dead protein is clearly getting abundantly phosphorylated by wild-type protein (Figure 2). But we could not detect an appreciable amount of CaMKII^KD^ integrated in CaMKII^F394BzF^ holoenzymes (Figure 3).

The blue arrows indicating bands in Figure 3 are very unclear – these images need to be processed so that the reader can actually see these bands.

We have processed these images now, in order to make the bands more visible.

On a related note, since your crosslinking is incomplete, the argument for inter-holoenzyme phosphorylation is weakened. If the crosslinking is improved and you can show similar rates of transphosphorylation, this will be significantly stronger data.

We agree that higher crosslinking efficacy would be more beneficial, which is why we did not base our conclusions solely on one experimental technique, but rather on several orthogonal methods (crosslinking mass-spectrometry, mass photometry, TIRF) to carefully examine the mixing of subunits upon activation. Each of these methods failed to detect subunits mixing, which taken together with the UV-induced crosslinking experiments makes a compelling argument for inter-holoenzyme phosphorylation as the main mechanism for spread of CaMKII phosphorylation, and therefore activity.

For the single-molecule experiments, single particles (ROIs) should be selected based on threshold, shape, and size using the General Analysis module in Nikon software, since that's the type of microscope you're using. You should set minimum and maximum thresholds to ensure that you're analyzing a similar population of particles. If you're unable to select single particles for analysis, then this is not single molecule resolution and the experiment should be repeated at lower concentrations or on cleaner coverslides (whatever the issue may be). The images shown in the main paper appear not to be well-resolved, but some images in the supplement do look well-resolved, this just needs to be analyzed properly and ensure that the microscope is properly aligned. If you're able to analyze the intensities of single particles, this should be plotted as a histogram. According to your model, the histogram of particles under basal conditions should correspond to a single holoenzyme, under activation conditions, the intensities should correspond to a double holoenzyme (as you see a small population of these in MP), and after EGTA treatment the intensities should again correspond to a single holoenzyme. The authors claim 80-90% labeling efficiency but should explain how this is determined – because if this is true I would expect homogenous particles that are quite bright (12-14 fluorophores) under basal conditions.

We are grateful for the suggestions for the image processing. We worked on this already quite a lot and continue to do so with new experiments. We have already thresholded the data carefully to achieve what the reviewer suggests- this explanation we now improved. The idea of single molecules is tricky. We don’t think we have single molecules (and we now removed reference to single molecules). We have groups of molecules.

We did some experiments at lower concentrations, and saw no differences. But we are almost at the low concentration limit in any case (55 nM).

As for plotting a histogram of particle intensities, we have tried to do this in the past. Again the problem is that we probably don’t have 12-mers and 24-mers, but rather groups of holoenyzmes. The key point is that the overlap of distinct coloured spots that we see with activity is reversibly lost. This point is quite decisive in our view, because the experiment (and therefore its interpretation) is very simple.

We have now improved our explanation of the labelling efficiency, please see around the line 757 in the manuscript.